# THE SELF-RE-WATERMARKING TRAP: FROM EXPLOIT TO RESILIENCE

**Vithurabiman Senthuran[1], Yong Xiang[1], Iynkaran Natgunanathan[1] & Uthayasanker Thayasivam [2]**
[1]Deakin University, [2]University of Moratuwa
{v.senthuran, yong.xiang, iynkaran.natgunanathan}@deakin.edu.au,
rtuthaya@cse.mrt.ac.lk

## ABSTRACT

Watermarking has been widely used for copyright protection of digital images. Deep learning-based (DL) watermarking systems have recently emerged as more effective than traditional methods, offering improved fidelity and resilience against attacks. Among the various threats to DL watermarking systems, self-re-watermarking attacks represent a critical and underexplored challenge. In such attacks, the same encoder is maliciously reused to embed a new message into an already watermarked image. This process effectively prevents the original decoder from retrieving the original watermark without introducing perceptual artifacts. In this work, we make two key contributions. First, we introduce the self-re-watermarking threat model as a novel attack vector and demonstrate that existing state-of-the-art watermarking methods consistently fail under such attacks. Second, we develop a self-aware watermarking framework to defend against this threat. Our key insight for mitigating this risk is to limit the sensitivity of the watermarking models to the inputs, thereby resisting re-embedding of new watermarks. To achieve this, we propose a self-aware deep watermarking framework that extends Lipschitz constraints to the watermarking process, regulating encoder–decoder sensitivity in a principled manner. In addition, the framework incorporates re-watermarking adversarial training, which further constrains sensitivity to distortions arising from re-embedding. The proposed method provides theoretical bounds on message recoverability under malicious encoder based re-watermarking and demonstrates strong empirical robustness against diverse scenarios of re-watermarking attempts. Moreover, it maintains high visual fidelity and demonstrates competitive robustness against common image processing distortions compared to state-of-the-art watermarking methods. This work establishes a robust defense against both standard distortions and self-re-watermarking attacks. Code available at `https://github.com/SVithurabiman/SRW`.

## 1 INTRODUCTION

Digital image watermarking plays a crucial role in preserving ownership and copyright protection for visual content distributed across digital platforms (Jia et al., 2021; Tancik et al., 2020; Luo et al., 2024). While modern deep learning (DL) based watermarking methods often outperform classical methods in terms of robustness and imperceptibility, they still remain vulnerable to adversarial attacks (Wang et al., 2021; Kinakh et al., 2024). One such threat is the re-watermarking attack, where an adversary embeds a new watermark into an already watermarked image, potentially causing the respective decoder to recover the second watermark instead of the original one. This attack transfers the ownership to an attacker, effectively allowing them to claim the image as their own and actively undermining the credibility of the watermarking systems.

Re-watermarking attacks in image watermarking systems can be broadly categorized into two types: (1) cross-model re-watermarking, where a different watermarking model embeds a new message into an already watermarked image (Chen et al., 2024b; Padhi et al., 2024a); and (2) self-re-watermarking, where the same encoder is directly reused to embed a new message. In cross-model re-watermarking, different embedding patterns between models typically allow both the original and the new watermarks to be independently recovered. This makes such attacks detectable through

multi-decoder inconsistencies (Please refer to Appendix A for further details). In contrast, self-overwriting poses a more severe threat. Since the same encoder is applied to an already watermarked image using the identical learned embedding function, it results in the removal of the original message, rendering it irretrievable. Such an attack hijacks the model's own logic to overwrite ownership without leaving detectable artifacts.

To investigate the severity of this threat, we conducted an empirical study on state-of-the-art deep watermarking models in the literature (Huang et al., 2023; Zhu et al., 2018; Fernandez et al., 2022; Jia et al., 2021; Luo et al., 2024; Lu et al., 2025) and found that they consistently fail under self-overwriting attacks. This exposes a systemic vulnerability in current designs that highlights the need for deeper analysis. Most recent deep learning-based watermarking approaches predominantly leverage encoder-decoder architectures trained to optimize for imperceptibility and robustness against common image processing distortions. However, these methods implicitly assume single-use embedding and do not account for repeated watermarking. While prior works have studied adversarial robustness for watermarking systems in the context of copyright protection (Chen et al., 2024a; Padhi et al., 2024a; Singh et al., 2024; Liu et al., 2022), to the best of our knowledge, none address the self-re-watermarking attack wherein the encoder is reused maliciously on watermarked content to re-embed a new watermark. To defend against self-re-watermarking attacks, the system must detect unauthorized overwriting and reliably recover the original watermark, ensuring ownership cannot be hijacked. Addressing this gap is critical for maintaining the integrity and trustworthiness of watermarking systems.

This work focuses on defending against a white-box adversary who has full access to the watermarking model, as these models cannot be assumed to remain protected indefinitely. Such access may result from model leakage, sharing, or reverse engineering. This scenario is particularly concerning because image owners may continue using the watermarking system without realizing that the model has been compromised. Even if the leaked model is later discarded, previously watermarked images remain vulnerable. Our motivation is further supported by recent research on model extraction (Rakin et al., 2022; Hu & Pang, 2021) and real-world model leakage incidents, such as the LLaMA case (Vincent, 2023). To this end, we propose a self-aware deep watermarking system designed to recover original messages even under self-overwriting. Our key insight is to develop a proactive watermarking framework that leverages a Lipschitz-constrained architecture (Cisse et al., 2017) to ensure reliable recovery of the original message even from overwritten images. We demonstrate that integrating these constraints directly into watermarking architectures offers a practical and effective approach to enhancing robustness against self-overwriting. This constraint ensures robustness to structured distortions introduced by re-embedding. To comprehensively defend against white-box adversaries capable of crafting targeted perturbations to mislead the decoder, we also employ adversarial training restricted to small pixel-level changes. By jointly enforcing bounded sensitivity and adversarial robustness in the system, our framework effectively resists both self-overwriting and norm-bounded re-watermarking attacks, preserving message fidelity and invalidating unauthorized re-use of the model. Although Lipschitz constraints have been studied previously in deep learning, integrating them into a watermarking system limits the encoder's capacity to preserve visual fidelity and prevents the decoder from maintaining sufficient robustness. To address this, we implement adaptive loss-weighting strategies that simultaneously preserve fidelity, enhance robustness against image-processing attacks, and protect against self-re-watermarking.

The major contributions of this study can be summarized as follows :

- Introduces **the self-re-watermarking threat model** in image watermarking, where the encoder is reused to embed a new message into an already watermarked image. Systematic experiments show that existing deep watermarking systems fail under this attack, revealing a significant vulnerability.

- Presents a **novel watermarking framework** built on a Lipschitz-constrained encoder–decoder architecture, enhanced with re-watermarking adversarial training and adaptive loss weighting. This design jointly optimizes fidelity and robustness, addressing both overwrite attacks and common image-processing distortions within a unified objective.

- **Formally analyzes** the system's bit-error rate under self-re-watermarking, using the same encoder. The work offers a theoretical bound for this attack class and complements it with extensive empirical evaluations to assess the system's robustness under overwriting and various image-processing attacks.

## 2 RELATED WORK

To contextualize our study, this section reviews two core areas: advances in deep learning–based watermarking and the evolving adversarial threats and countermeasures.

### 2.1 DL BASED IMAGE WATERMARKING

Deep learning has become central to image watermarking, enabling models to balance imperceptibility and robustness. Early work by Baluja et al. (Baluja, 2017) proved the feasibility of DL-based steganography, while HiDDeN (Zhu et al., 2018) introduced differentiable noise layers to simulate distortions such as cropping and compression. To address non-differentiable or unknown distortions, Luo et al. (2020) proposed a distortion-agnostic framework with adversarial training, and MBRS (Jia et al., 2021) further improved robustness to JPEG by mixing real and simulated codecs. Other advances include ARWGAN (Huang et al., 2023), which applied attention-based fusion, and Fernandez et al. (Fernandez et al., 2022), who used self-supervised learning with DINO (Caron et al., 2021) to target semantically meaningful regions, although such methods remain vulnerable to cropping. Transformer-based designs such as WFormer (Luo et al., 2024) and security-focused schemes like GANMarked (Singh et al., 2024) improved robustness and key protection, yet struggled against forgery or adaptive attacks. Recently, Lu et al. (2025) developed VINE to address vulnerabilities in watermarking against large-scale text-to-image models.

### 2.2 ADVERSARIAL ATTACKS IN DL BASED IMAGE WATERMARKING

DL watermarking faces adversarial threats beyond removal attacks (Zhao et al., 2024; An et al., 2024). A critical yet underexplored risk is *self-re-watermarking*, where an attacker reuses the encoder to embed a conflicting message into a watermarked image, creating false ownership claims. Kinakh et al. (2024) showed that self-supervised methods are prone to unauthorized transfer, while forgery-based strategies (Hu et al., 2025) can fabricate counterfeit ownership. These studies demonstrate how adversarial pressure on watermarking systems is expanding in sophistication.

Some studies have focused on defending against particular classes of adversarial attacks. For instance, diffusion-based schemes (Zhu et al., 2024) embed adversarial watermarks to obstruct generative imitations. Recent dual watermarking efforts (Padhi et al., 2024b) attempt to resist model style-transfer attacks. Other approaches target overwriting, such as high-frequency embedding (Chen et al., 2024b) or dual-watermarking (Padhi et al., 2024a), but their scope is narrow. Overall, robust countermeasures against self-re-watermarking remain absent. Building on this gap, we propose a framework that reduces model sensitivity to input changes, preserving robustness to standard distortions while resisting adversarial overwriting.

## 3 THREAT MODEL: SELF-RE-WATERMARKING ADVERSARY

In this section, we first define the problem setup, the adversary's capabilities and objectives, and then proceed to the attack mechanisms.

### 3.1 PROBLEM SETUP AND NOTATIONS

This work considers a white-box adversary $O$ that has full access to the encoder $E$ and decoder $D$. The Encoder $E$ takes in normalized real image $x \in \mathcal{X}$ and bipolar messages $m \in \widetilde{\mathcal{M}}$ to produce watermarked images $x_w \in \mathcal{X}$, where $\mathcal{X} \subset [-1,1]^{H \times W \times 3}$ and $\widetilde{\mathcal{M}} = \{-1,1\}^L$ Meanwhile, the Decoder $D$ takes in $x_w \in \mathcal{X}$ and produces logits $Z \in \mathbb{R}^L$ which can be mapped to bit values.

### 3.2 ADVERSARIAL CAPABILITIES

We consider a **white-box adversary** with full access to the model parameters and training procedure, and resources to develop and train watermarking models of comparable complexity. Such an adversary can launch three types of attacks. First, through **Encoder-Based Self-Re-Watermarking** ($O_{\text{SRW}}$), the adversary can directly reuse the encoder to embed a new message into an already wa-

termarked image. Second, using **Gradient-Based Adversarial Attack** ($O_{\text{GBA}}$), the adversary leverages the decoder's gradients to generate a perturbation bounded by a maximum allowable pixel change that, when added to the watermarked image, compels the decoder to output a target message. Finally, with **Model Replication-Based Overwrite Attack** ($O_{\text{MR}}$), the adversary exploits knowledge of the training algorithm and loss functions to train a surrogate watermarking model, enabling them to embed a new watermark and overwrite the original one without requiring access to the original model parameters or training data.

### 3.3 ADVERSARIAL OBJECTIVES

In the context of these attack strategies, adversaries generally pursue two primary objectives. First, they aim to **overwrite the original message**, such that the decoder produces a target message. Second, these modifications are often constrained by **perceptual fidelity**, requiring that the re-watermarked image remains perceptually similar to the original watermarked image.

### 3.4 THE SELF OVERWRITING ATTACK

As the adversary has access to the encoder model and its parameters, they can directly embed an adversarial watermark $m' \in \widetilde{\mathcal{M}}$ into an already watermarked image $x_w$, resulting in a re-watermarked image $x_{w'} \in \mathcal{X}$. Formally, the process can be expressed as

$$x_{w'} = O_{SRW}(E(x,m); m') = E(x_w, m'), \quad \text{where } m' \neq m. \tag{1}$$

### 3.5 GRADIENT BASED ADVERSARIAL OVERWRITING

Beyond maliciously reusing the encoder, we also consider a powerful adversary who is capable of creating a subtle perturbation $\psi$ to fool the decoder while maintaining high visual fidelity. To achieve this, we formulate this attack as an iterative Projected Gradient Descent (PGD) optimization (Madry et al., 2017). At each iteration, the adversary updates the adversarial image, $x_{\text{adv}}$, to minimize the decoder's binary cross-entropy with the target message $m_g \in \mathcal{M}$, projecting the perturbation onto an $\ell_\infty$-norm ball of radius $\epsilon$ and clipping to valid pixel ranges. The detailed algorithmic procedure is given in **Algorithm 1**. Formally, this attack can be described as follows

$$x_{adv} = O_{GBA}(E(x,m); m_g) = E(x,m) + \psi \tag{2}$$

---

**Algorithm 1** PGD Self-Overwrite Attack

---

**Require:**
 1: Watermarked image $x_w \in [-1,1]^{B \times C \times H \times W}$, Target message bits $m_g \in \{0,1\}^{B \times L}$, Decoder function $D$, Maximum perturbation $\epsilon$, Step size $\alpha$, Number of iterations $T$
**Ensure:** Adversarial image $x_{adv}$
 2: Initialize $x_{adv} \leftarrow x_w$
 3: **for** $i = 1$ **to** $T$ **do**
 4:     Compute logits: $Z \leftarrow D(x_{adv})$
 5:     Compute loss: $\mathcal{L} = \text{BCEWithLogitsLoss}(Z, m_g)$
 6:     Compute gradient: $g \leftarrow \nabla_{x_{adv}} \mathcal{L}$
 7:     Gradient descent step with sign: $x_{adv} \leftarrow x_{adv} - \alpha \cdot \text{sign}(g)$
 8:     Project perturbation back to $\ell_\infty$ ball: $\delta \leftarrow \text{clip}(x_{adv} - x_w, -\epsilon, \epsilon)$
 9:     Clamp to valid image range: $x_{adv} \leftarrow \text{clip}(x_w + \delta, -1, 1)$
10: **end for**
11: **return** $x_{adv}$

---

## 4 PROPOSED METHODOLOGY

This section proposes a principled approach to resist self-re-watermarking attacks in watermarking systems by jointly optimizing fidelity, nominal recovery, and robustness.

### 4.1 MODEL ARCHITECTURE

The proposed architecture is designed to be sufficiently expressive to embed messages within images while preserving their fidelity. Furthermore, it is carefully structured to ensure robustness against self-overwriting attacks through Re-Watermarking Adversarial Training, thereby enabling reliable message extraction, even after self-re-watermarking. The proposed architecture comprises the following key components:

**Encoder:** We adopt a U-Net architecture (Ronneberger et al., 2015), a widely used design in watermarking algorithms, to evaluate how bounded sensitivity can be adapted to such architectures. To support multi-scale feature extraction, we incorporate an auxiliary ResNet-50 backbone (He et al., 2016). The input to the encoder is constructed by concatenating the cover image $x \in \mathbb{R}^{3 \times H \times W}$ with the message $m \in \{-1, 1\}^L$. Prior to concatenation, the message undergoes spatial expansion through spectrally normalized linear layers. This results in a 4-channel tensor input.

The encoder consists of four downsampling blocks augmented with skip connections from intermediate ResNet layers. A bottleneck with spectral and group normalization connects to four upsampling blocks with skip connections and ReLU activations. A final spectrally normalized $1 \times 1$ convolution produces a residual image, which is added to the input image.

**Decoder:** The decoder is a convolutional neural network that recovers the embedded message from the watermarked image. Each convolutional block comprises a spectrally normalized convolutional layer with kernel size $3 \times 3$, followed by group normalization (with 4 groups) and a ReLU activation. Residual connection between the blocks, enhancing gradient flow and feature reuse. The final fully connected layer produces the final message logits.

**Noise Model:** To simulate real-world distortions, we employ a differentiable noise model composed of common image perturbations. At each training iteration, one perturbation is randomly sampled from a pool that includes JPEG compression, Gaussian blur, dropout, cropout, cropping, horizontal, vertical flips, scaling, and rotation. Each selected operation is applied with randomly sampled parameters within plausible ranges.

**Post-Processing Module:** During inference, the watermarked image undergoes Gaussian blurring followed by suppression of low-magnitude values to enhance the visual fidelity.

### 4.2 TRAINING OBJECTIVE

The training objective is designed to meet three goals: preserving image fidelity, ensuring reliable nominal recovery of the embedded message, and maintaining robustness against self-overwriting attacks. To achieve this, the system optimizes a composite loss with three components. First, the fidelity loss enforces the watermarked image to remain visually consistent with the cover image. It integrates both mean squared error and a perceptual similarity term, measured via LPIPS (Zhang et al., 2018), given by:

$$\mathcal{L}_{\text{fid}} = \text{MSE}(x, x_w) + \lambda_{\text{lpips}} \cdot \text{LPIPS}(x, x_w) \tag{3}$$

Second, the nominal recovery loss ensures reliable message extraction under benign conditions:

$$\mathcal{L}_{\text{rec}} = \text{BCE}(D(x_w), \phi(m)) \tag{4}$$

where the function $\phi$ maps $m$ from its bipolar form to standard bit values. Third, the robustness loss is designed to enhance resilience against overwriting attacks. Specifically, it penalizes decoding errors when the system is confronted with re-watermarked images generated through malicious encoder reuse, as well as adversarially perturbed inputs obtained via gradient-based optimization. It is given by:

$$\mathcal{L}_{\text{rec}} = \text{BCE}(D(x_w), \phi(m)) \tag{5}$$

Thus, the full optimization problem can be formulated as:

$$\min_{\theta_E, \theta_D} \mathbb{E}_{x, m, m'} \left[ \lambda_{\text{fid}} \cdot \mathcal{L}_{\text{fid}} + \lambda_{\text{rec}} \cdot \mathcal{L}_{\text{rec}} + \lambda_{\text{rob}} \cdot \mathcal{L}_{\text{rob}} \right] \tag{6}$$

where $\theta_E$ and $\theta_D$ are the parameters of $E$ and $D$ respectively. $\lambda_{\text{fid}}$, $\lambda_{\text{rec}}$, and $\lambda_{\text{rob}}$ can be adaptively changed using the nominal bit recovery and the bit recovery of the original message after adversarial training as per Algorithm 2 in Appendix C.

## 4.3 TRAINING PIPELINE

In this work, we construct a training pipeline that integrates noise modeling and adversarial simulations. During training, the encoder embeds a binary message into the cover image to produce a watermarked image. This image is then optionally passed through a noise model simulating common distortions described under Subsection 4.1. The decoder attempts to recover the embedded message from the (possibly distorted) watermarked image. Figure 1 illustrates our training pipeline.

A PGD-based adversarial overwriting scenario is also simulated during training. This forces the model to learn robustness against adaptive gradient-based attacks. Additionally, to further enhance resilience, a self-overwriting scenario is simulated by feeding the watermarked image back into the encoder to mimic an adversary attempting to re-embed a new message on top of the existing watermark. This encourages the model to maintain watermark integrity under repeated embedding attempts. Together, these training strategies ensure robustness against both gradient-based and self-overwriting adversarial manipulations. While these mechanisms offer strong empirical protection, the following section will formally analyze the system's robustness under malicious encoder reuse.

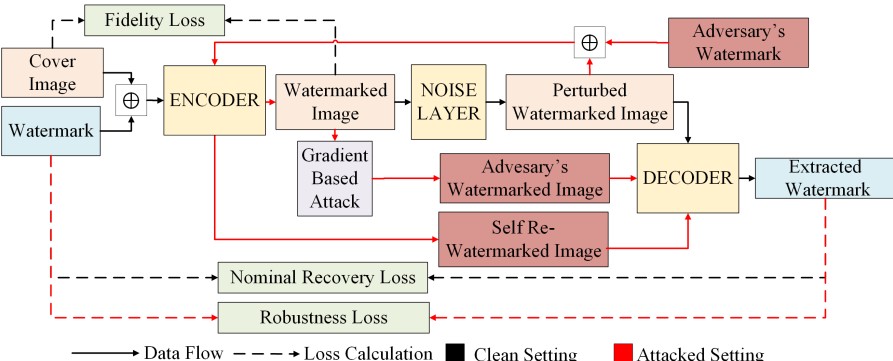

Figure 1: Overview of the training pipeline for the proposed system, illustrating both the standard watermarking process (black arrows) and the adversarial training loop (red arrows) used to ensure robustness against attacks.

## 4.4 LIPSCHITZ CONSTRAINTS AND ASSUMPTIONS

1. **Decoder Lipschitzness** There exists an upper bound $K_D$ such that for all images $x_1, x_2 \in \mathcal{X}$,

$$\|D(x_1) - D(x_2)\|_\infty \leq K_D \|x_1 - x_2\|_\infty. \tag{7}$$

In practice, $K_D$ can be a global constant (conservative) or a *data-dependent local estimate* measured along the path from watermarked image $x_w$ to re-watermarked image $x_{w'}$:

$$K_{D,\text{loc}} := \frac{\|D(x_{w'}) - D(x_w)\|_\infty}{\|x_{w'} - x_w\|_\infty}. \tag{8}$$

2. **Positive clean margin.** The minimum signed margin across all images and bits, which guarantees that every bit is correctly decoded in the absence of an overwrite:

$$\Delta_{\min} := \inf_{x,m,i} \Delta_i(x, m) > 0 \qquad where \qquad \Delta_i(x, m) := m_i D_i\big(E(x,m)\big) \tag{9}$$

This quantity measures the worst-case "safety buffer" for the decoder logits, i.e., the smallest distance of any bit logit from zero under clean conditions.

These assumptions facilitate a feasible robustness analysis. In practice, they are supported by architectural constraints and training procedures. Supporting empirical evaluation of these quantities is detailed in Appendix B.4.

## 4.5 THEORETICAL ANALYSIS

This subsection analyzes the decoder's robustness to self-re-watermarking, deriving an error bound and a theorem that upper-bounds the bit error rate (BER) between the decoded messages after re-watermarking and the original messages. Formal proofs are provided in **Appendix B**. By defining the nominal decoder error as $\varepsilon_{\text{rec}} = \sup_{x,m} \frac{1}{L}\sum_{i=1}^{L} \mathbf{1}\big(\text{sign}(D_i(x,m)) \neq m_i\big)$, the distortion introduced due to overwriting as $\delta_\infty = \|x_{w'} - x_w\|_\infty$, the standard sign function as $\text{sign}(\cdot)$, and the indicator function as $\mathbf{1}(\text{condition})$, we can state the following theorem.

**Theorem 1** (BER upper bound). *For a given triplet $(x, m, m')$ with overwrite $x_{w'}$, the bit error rate satisfies*

$$\text{BER}(x, m, m') \leq \frac{1}{L}\sum_{i=1}^{L} \mathbf{1}\big(\Delta_i(x,m) \leq K_D \delta_\infty\big) + \varepsilon_{\text{rec}}. \tag{10}$$

*In particular, if $K_D\delta_\infty < \Delta_{\min}$, the overwriting process does not flip the bit, therefore*

$$\text{BER}(x, m, m') \leq \varepsilon_{\text{rec}}. \tag{11}$$

**Corollary 1** (Local, data-dependent tightening). *Replacing $K_D$ by the local, attack-path constant $K_{D,\text{loc}}$ yields the tighter bound*

$$\text{BER}(x, m, m') \leq \frac{1}{L}\sum_{i=1}^{L} \mathbf{1}\big(\Delta_i(x,m) \leq K_{D,\text{loc}}\,\delta_\infty\big) + \varepsilon_{\text{rec}}. \tag{12}$$

**Corollary 2** (Perfect recovery under margin condition). *If $\varepsilon_{\text{rec}} = 0$ and $K_D\,\delta_\infty < \Delta_{\min}$, then no bits flip under overwrite, and hence*

$$\text{BER}(x, m, m') = 0, \quad \forall(x, m, m'). \tag{13}$$

## 5 EXPERIMENTAL SETTING

This section outlines the datasets, evaluation metrics, baselines, and implementation details used to validate the effectiveness of the proposed method. In our experiments, we consider the following seven state-of-the-art studies: dwtDctSvd (Navas et al., 2008), HiDDeN (Zhu et al., 2018), MBRS (Jia et al., 2021), SSL (Fernandez et al., 2022), ARWGAN Huang et al. (2023),WFormer (Luo et al., 2024), and VINE(Lu et al., 2025)

### 5.1 TRAINING SETTING

We use a subset of the COCO dataset Lin et al. (2014) consisting of 20,000 training, 1,000 validation, and 3,000 testing images. All RGB images are resized to $128 \times 128$ pixels and normalized with mean [0.5, 0.5, 0.5] and standard deviation [0.5, 0.5, 0.5]. Binary messages of length $L = 30$ bits are randomly sampled for watermarks. We set $\lambda_{lpips}$ as 0.5. Training and experiments were conducted on a dual-socket Intel Xeon E5-2670 system and RTX A4000 GPU. The Lipschitz constraint was enforced by applying spectral normalization to all convolutional and linear layers in the models.

### 5.2 EVALUATION METRICS

The performance of the proposed watermarking method is evaluated in terms of imperceptibility, reflecting the preservation of visual quality, and robustness, indicating the resilience of the embedded watermark to attacks and distortions. To assess imperceptibility, we report Peak Signal-to-Noise Ratio (PSNR) and Structural Similarity Index Measure (SSIM) between cover and watermarked images, where higher values indicate better visual quality, and a PSNR above 30dB is generally considered to reflect acceptable imperceptibility Zhang et al. (2024); Subhedar & Mankar (2020). For robustness evaluation, we measure three bit-accuracy metrics, computed per image and averaged over the test set: $\text{ACC}_{\text{clean}}$ evaluates message recovery under normal, non-adversarial conditions; $\text{ACC}_{\text{adv}}$ measures the accuracy between the decoder output and the adversarial target message after attacks such as self-overwriting or gradient-based perturbations; and $\text{ACC}_{\text{orig}}$ quantifies the similarity between the decoded message post-attack and the originally embedded watermark, indicating how well the original watermark withstands adversarial manipulations. Higher values of $\text{ACC}_{\text{clean}}$, $\text{ACC}_{\text{adv}}$, and $\text{ACC}_{\text{orig}}$ indicate better message recovery.

## 5.3 SELF-RE-WATERMARKING ATTACK ON EXISTING WORK

In this subsection, we investigate the vulnerability of existing deep learning-based watermarking models to *self-re-watermarking attacks*. To systematically evaluate the robustness of watermarking models under self-re-watermarking, we design a controlled experimental protocol consisting of three key scenarios as described under Subsection 5.2

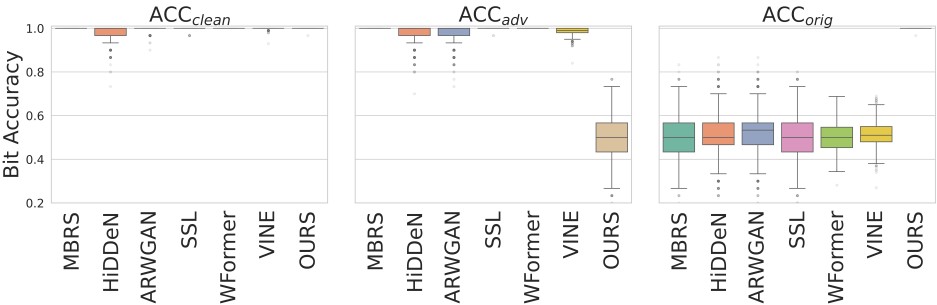

Figure 2: Bit accuracy under self-re-watermarking attacks using their respective encoders

To visually compare our proposed model with other learning based SOTA models under self-re-watermarking attacks, we present scenario-wise box plots in Figure 2. The figure illustrates the distribution of bit accuracies over the test set, highlighting differences in overwrite robustness. High bit accuracy in the first two scenarios confirms effective watermark embedding and retrieval, while a low bit accuracy in the third scenario indicates successful erasure of the original watermark under self-overwrite attacks. Figure 2 shows that all SOTA models fail under malicious encoder reuse, whereas the proposed model withstands the attack and successfully recovers the original watermark even after an adversary attempts to re-embed a new one. Moreover, as shown in **Appendix I**, the re-watermarking process in our model visibly distorts the resulting image, preventing an adversary from gaining any advantage through iterative re-watermarking. Quantitatively, the average PSNR and SSIM between the watermarked and re-watermarked images are 10.21 dB and 0.66, respectively, indicating severe degradation. Figure 3 provides an illustration of the cover, watermarked, and re-watermarked images.

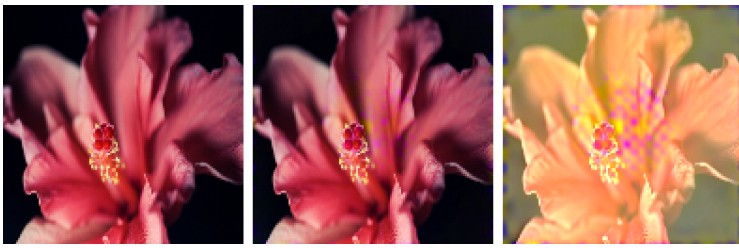

Figure 3: Cover, watermarked, and re-watermarked images generated by our model, respectively.

## 5.4 EMPIRICAL EVALUATION OF THE THEORETICAL BOUND

This section empirically analyzes the theoretical bound of the system when re-watermarked using our Encoder, as this represents the most challenging attack conditions. As per the bound, we compare the minimum per-bit clean margin $\Delta_{\min}$ to the empirical Lipschitz-based theoretical lower bound $K_{D,loc}\delta_{\infty}$ to assess how well the bound reflects real-world behavior. It should be noted that this section analyzes only the most vulnerable bit, rather than all embedded bits. This evaluation examines both the theoretical and practical robustness in the worst-case scenario, making the resulting bound conservative. Figure 4a illustrates the relationship between the minimum-margin bit $\Delta_{min}$ and the per-image overwrite bound $K_{D,loc}\delta_{\infty}$ across 3,000 images, along with the observed bit flips. Green points indicate bits correctly decoded after overwrite, while red points indicate flipped bits. As expected, all points above the line $K_{D,loc}\delta_{\infty} = \Delta_{min}$ remain green, confirming that bits with

margins exceeding the bound are reliably robust. Below this threshold, we observe a mixture of green and red points.

Figure 4b isolates only the flipped bits, which all fall below the $K_{D,loc}\delta_\infty = \Delta_{min}$ line, i.e., inside the region where the theorem predicts potential vulnerability. This demonstrates that the theoretical bound provides a conservative, yet informative, necessary condition for bit flips. The analysis shows that $K_{D,loc}\delta_\infty$ provides a conservative estimate for bit flips, while not all bits below the threshold actually flip, all observed flips occur within this region, which is consistent with the theoretical bound. While the bound predicts potential vulnerability, the majority of these bits still survive, showing that the theoretical estimate is conservative yet valid. These observations confirm that the bound provides a useful, conservative estimate of robustness while the trained decoder shows added resilience against self-overwriting.

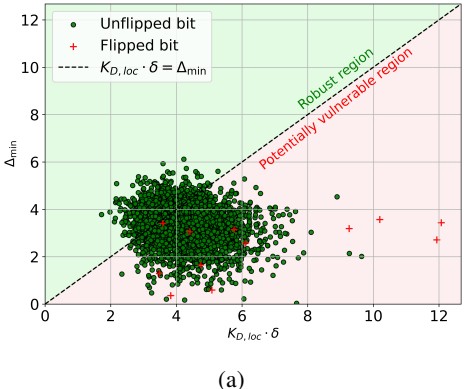
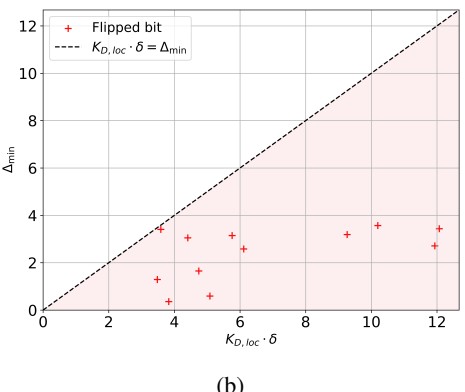

(a)             (b)

Figure 4: Scatter plot illustrating the relationship between $K_D \cdot \delta$ and the minimum distance $\Delta_{\mathrm{min}}$ for watermark embedding. Each point represents an image sample. Green points represent unflipped bits, and red points represent flipped bits. The dashed line indicates the theoretical bound.

## 5.5 Adversarial Attack Evaluation

This subsection evaluates the robustness of the system under advanced adversarial scenarios. As discussed in Section 3.2, we assume an adversary capable of executing a PGD-based self-overwriting attack, implemented as detailed in Algorithm 1. Unlike malicious encoder reuse, the PGD attack iteratively applies small perturbations to mislead the decoder to output the target watermark.

We simulate this attack using two configurations. The first, termed **PGD$_{\textbf{moderate}}$**, uses $\epsilon = 0.03$, $\alpha = 0.007$, and 50 iterations. The second, **PGD$_{\textbf{strong}}$**, is more aggressive, with $\epsilon = 0.04$, $\alpha = 0.01$, and 100 iterations. Figures 5a and 5b illustrate the effectiveness of these attacks across several state-of-the-art models, and demonstrate the robustness of the proposed model in preserving the watermark under both moderate and strong attack settings. The numerical values are presented in Table 1. Further analysis of the perturbation budget ($\epsilon$), as detailed in **Appendix D.1**, reveals that the model's performance starts to deteriorate as the perturbation budget increases. However, when this happens, the resulting image quality drops below 30dB, causing noticeable degradation, which makes the resulting image less valuable for any potential adversary.

In addition, we consider an adversary who constructs a watermarking system using a similar architecture to ours with a different dataset. In the first setting, the adversary uses the same losses as ours, denoted as *Baseline Adversarial Model (BAM)*. In the second, the adversary omits the robustness loss, optimizing only for imperceptibility and accuracy, denoted as *Ablated Model (AM)*. Visual quality and watermark extraction results of *BAM* and *AM* are reported in **Appendix. D.2**. The ability of our decoder to recover the original watermark after attacks by these adversarial models demonstrates remarkable performance. Specifically, it achieves 100% accuracy against BAM, irrespective of whether a post-processing model is used, and against AM with post-processing, while achieving 99% accuracy against AM without post-processing. This indicates that the proposed method consistently resists overwrite attempts. Furthermore, the encoder perceptually degrades re-watermarked images, preventing black box query-based API attacks from producing meaningful outputs.

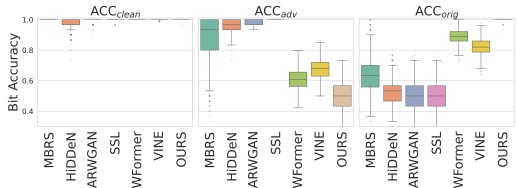 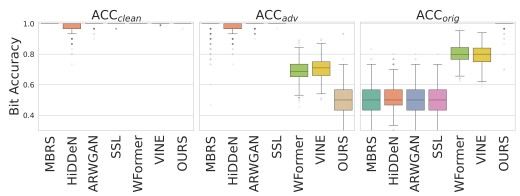

(a) Bit accuracy under PGD$_{moderate}$ attack ($\epsilon = 0.03$, $\alpha = 0.007$, 50 iterations).

(b) Bit accuracy under PGD$_{strong}$ attack ($\epsilon = 0.04$, $\alpha = 0.01$, 100 iterations).

Figure 5: Comparison of bit accuracy under moderate and strong PGD attacks. The proposed model shows higher robustness compared to SOTA methods.

## 5.6 ROBUSTNESS TO NOISE AND DISTORTIONS

While our main goal is to defend against self-re-watermarking attacks, the watermark must also remain retrievable under common image distortions. Therefore, we evaluated the proposed model's robustness against a standard set of such perturbations. As shown in Table 1, the proposed model maintains exceptionally high robustness across all tested distortions. It achieves near-perfect bit recovery under Gaussian blur (99.66%), dropout (98.90%), cropout (98.14%), cropping (99.85%), and JPEG compression (95.06%). Visual metrics such as PSNR (34.03 dB) and SSIM (0.97) confirm that the watermarked images are perceptually closer to the cover images. Further analysis on the robustness to different pixel-wise and geometric distortions is available in **Appendix F.1**. These results indicate that the model's watermark embedding is robust to diverse transformations.

Table 1: Comparison of the proposed model with SOTA baselines across visual quality, robustness to image processing, and robustness to adversarial overwrite attacks.

| Studies | Visual Quality | | ACC$_{clean}$ (%) | | | | | ACC$_{orig}$ (%) | | |
|---------|------------|------|------|----------|---------|---------|--------|-----------|----------|--------|
| | PSNR (dB) | SSIM | JPEG (50) | Gaussian Blur (2.0) | Dropout (30%) | Cropout (30%) | Crop (3.5%) | Self Re-embed | PGD Moderate | PGD Strong |
| dwtDctSvd | 28.57 | 0.94 | **99.97** | 99.41 | 54.36 | 85.40 | 51.29 | 50.00 | N/A | N/A |
| Hidden | 33.55 | 0.92 | 63.00 | 96.00 | 93.00 | 94.00 | 88.00 | 51.29 | 52.03 | 51.45 |
| MBRS | 35.84 | 0.89 | 91.97 | **100.00** | 99.96 | 99.98 | 92.68 | 50.34 | 63.51 | 51.26 |
| SSLW | 33.10 | 0.94 | 83.01 | 98.96 | 88.11 | 79.66 | 50.73 | 49.90 | 49.81 | 49.81 |
| ARWGAN | 35.87 | 0.96 | 93.98 | 99.99 | **100.00** | 99.82 | 98.17 | 51.94 | 50.68 | 50.73 |
| WFORMER | 33.50 | 0.91 | 99.14 | **100.00** | **100.00** | **100.00** | 98.70 | 50.02 | 88.64 | 80.15 |
| VINE | **37.07** | **0.99** | 99.97 | 99.84 | 87.63 | 99.99 | 52.24 | 51.20 | 82.00 | 79.41 |
| Proposed | 34.03 | 0.97 | 95.06 | 99.66 | 98.90 | 98.14 | **99.85** | **100.00** | **99.95** | **99.37** |

## 6 CONCLUSION

This work introduces the self-re-watermarking threat model, an overlooked but critical vulnerability in image watermarking systems, where adversaries can reuse the encoder to overwrite embedded watermarks without perceptible changes. We demonstrated that existing watermarking methods are highly vulnerable to such an attack. To mitigate this attack, we introduce a robust watermarking framework that combines architectures with bounded sensitivity with re-watermarking adversarial training. Further, this work formally analyzes watermark recoverability and exhibits strong empirical resilience against both self-re-watermarking and norm-bounded re-watermarking attacks. In addition, it maintains high visual fidelity and robustness to standard pixel-wise and geometric distortions. A limitation of the current approach is that it focuses solely on self-re-watermarking attacks. Extending our approach to defend against different classes of adversarial attacks concurrently is a key direction for future research.

## ACKNOWLEDGMENTS

This work was supported in part by Ausintelli Technology Pty Ltd.

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

## APPENDIX

## A   ANALYSIS ON RE-WATERMARKING SCENARIOS

As outlined in the Introduction, re-watermarking can be divided into cross-model overwriting and self-re-watermarking. This section evaluates the effect of re-watermarking on four different baselines. Figure 6a shows $Acc_{orig}$, reflecting the ability of each decoder to recover the original watermark after re-watermarking across the four baseline models. The diagonal accuracies, representing the self-re-watermarking scenario, are noticeably low, with an average $Acc_{orig}$ of approximately 50%, which corresponds to random guessing. As shown in the figure, this behavior is consistent across all models. In contrast, in most cross-model cases, the original message can still be recovered by the respective decoders. In addition, Figure 6b illustrates the average PSNR between watermarked and re-watermarked images. The PSNR values, all above 30dB, indicate that the degradation is imperceptible. This further underscores the seriousness of self-re-watermarking attacks, as the modifications are visually unnoticeable and effectively prevent the original watermark from being recovered.

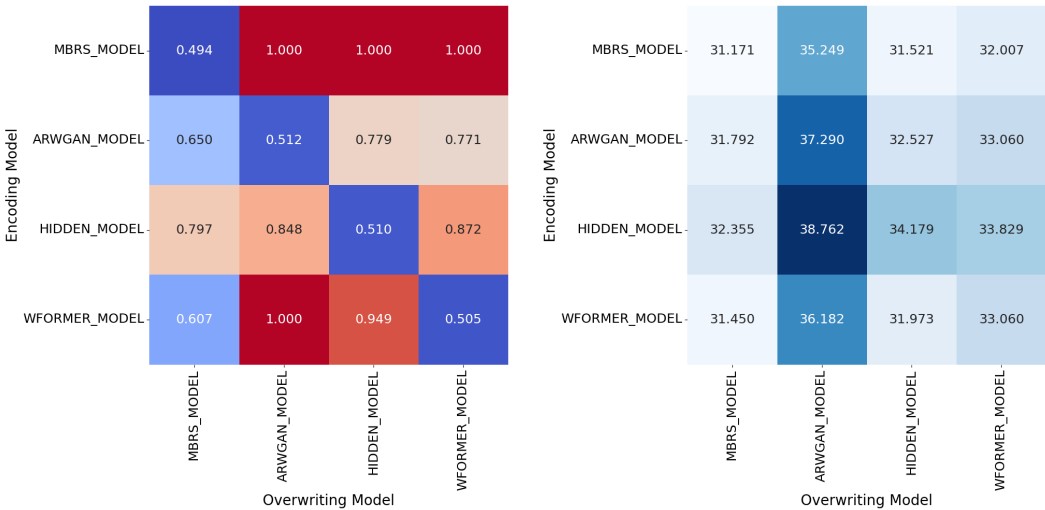

(a) Average $Acc_{orig}$ of watermarks decoded from re-watermarked images

(b) Average PSNR (dB) between watermarked and re-watermarked images

Figure 6: Analysis of Overwriting Scenarios.

## B   PROOFS OF THEORETICAL RESULTS

### B.1   PROBLEM SETUP AND NOTATION

Let:

- $\mathcal{X} \subset [-1, 1]^{H \times W \times 3}$ be the space of normalized RGB images.
- $\mathcal{M} = \{0, 1\}^L$ the binary message space and $\widetilde{\mathcal{M}} = \{-1, 1\}^L$ its bipolar version.
- $\mathbf{1}(\text{condition})$ denote the indicator function, equal to 1 if the condition is true and 0 otherwise.
- $\|\cdot\|_\infty$ denote the $\ell_\infty$ norm on images and vectors; unless otherwise stated, norms are $\ell_\infty$.
- $m \in \widetilde{\mathcal{M}}$ the original watermark and $m' \in \mathcal{M}$ the adversary's target watermark.
- $x \in \mathcal{X}$ the clean input image; $x_w \in \mathcal{X}$ the watermarked image with $m$; $x_{w'} \in \mathcal{X}$ the overwritten image.

**Encoder ($E$).**

$$E : \mathcal{X} \times \widetilde{\mathcal{M}} \to \mathcal{X}, \qquad x_w = E(x, m). \tag{14}$$

**Decoder ($D$).**

$$D : \mathcal{X} \to \mathbb{R}^L, \qquad \tilde{m}_i(x) = \text{sign}\big(D_i(x)\big) \in \{-1, 1\}. \tag{15}$$

(If a binary output is needed, use $\hat{m}_i(x) = \frac{1}{2}(\tilde{m}_i(x) + 1) \in \{0, 1\}$.)

**Overwrite distortion.**

$$\delta_\infty \;=\; \|x_{w'} - x_w\|_\infty \tag{16}$$

**Clean logits and margins.**

$$\ell_i(x, m) := D_i\big(E(x, m)\big), \qquad \Delta_i(x, m) := m_i \, \ell_i(x, m) \tag{17}$$

Thus $\Delta_i > 0$ means bit $i$ is correctly signed on $x_w$ with (signed) margin $\Delta_i$.

**Nominal decoder error.**

$$\varepsilon_{\text{rec}} \;=\; \sup_{x, m} \frac{1}{L} \sum_{i=1}^{L} \mathbf{1}\big(\text{sign}(\ell_i(x, m)) \neq m_i\big) \tag{18}$$

**Self-Overwriting Attack** The adversary can perform a *self-re-watermarking attack* $\mathcal{O}_{SRW}$ by overwriting an already watermarked image to produce $x'_w$:

$$x_{w'} = O_{SRW}(E(x, m); m') = E(x_w, m'), \quad \text{where } m' \neq \phi(m) \tag{19}$$

## B.2 Lipschitz Constraints and Assumptions

1. **Decoder Lipschitzness (analysis norm $\ell_\infty$).** There exists an upper bound $K_D$ such that for all $x_1, x_2 \in \mathcal{X}$,

$$\|D(x_1) - D(x_2)\|_\infty \;\leq\; K_D \, \|x_1 - x_2\|_\infty \tag{20}$$

   In practice, $K_D^\infty$ can be a global constant (conservative) or a *data-dependent local estimate* measured along the path from $x_w$ to $x_{w'}$:

$$K_{D,\text{loc}} \;:=\; \frac{\|D(x_{w'}) - D(x_w)\|_\infty}{\|x_{w'} - x_w\|_\infty} \tag{21}$$

2. **Positive clean margin.** The minimum signed margin across all images and bits, which guarantees that every bit is correctly decoded in the absence of an overwrite:

$$\Delta_{\min} \;:=\; \inf_{x, m, i} \Delta_i(x, m) \;>\; 0 \tag{22}$$

   This quantity measures the worst-case "safety buffer" for the decoder logits, i.e., the smallest distance of any bit logit from zero under clean conditions.

In our watermarking framework, the assumptions of decoder Lipschitzness and positive clean margins are incorporated and empirically enforced through architectural and training design. Spectral normalization in all convolutional and linear layers of the models constrains the layer-wise operator norms, effectively bounding the decoder's sensitivity to input changes and supporting the Lipschitz assumption. Positive clean margins are encouraged by the binary cross-entropy loss for nominal recovery, adversarial robustness losses, and noise augmentations, which collectively push decoder logits away from zero under both clean and perturbed conditions. These mechanisms ensure that the assumptions hold empirically for the training and test distributions.

## B.3 THEORETICAL ANALYSIS

**Lemma 1** (Per-logit overwrite bound). *For any $(x, m, m')$ and $x_{w'} = O(E(x, m); m')$,*

$$\|D(x_{w'}) - D(x_w)\|_\infty \leq K_D \, \delta_\infty \tag{23}$$

*Consequently, for each bit $i$,*

$$\left| D_i(x_{w'}) - D_i(x_w) \right| \leq K_D \, \delta_\infty \tag{24}$$

*Proof.* As per assumption 1, the decoder $D$ is $K_D$-Lipschitz with respect to the $\ell_\infty$ norm. Then for any two inputs $x_1, x_2 \in \mathcal{X}$ we have

$$\|D(x_1) - D(x_2)\|_\infty \leq K_D \|x_1 - x_2\|_\infty$$

The overwrite distortion was defined as

$$\delta_\infty = \|x'_w - x_w\|_\infty$$

Therefore,

$$\|D(x'_w) - D(x_w)\|_\infty \leq K_D \delta_\infty$$

Since the $\ell_\infty$ norm of the decoder difference corresponds to the maximum per-bit logit deviation, this inequality implies that every logit changes by at most $K_D \delta$ under overwriting. $\square$

**Proposition 1** (Per-bit robust condition). *Let $\Delta_i = \Delta_i(x, m)$ be the clean margin of bit $i$ If*

$$\Delta_i > K_D \, \delta_\infty$$

*then bit $i$ cannot flip under the overwrite, i.e.*

$$\operatorname{sign}\left( D_i(x_{w'}) \right) = \operatorname{sign}\left( D_i(x_w) \right) = m_i$$

*Proof.* Write

$$D_i(x_{w'}) = D_i(x_w) + e_i$$

By Lemma 1, the perturbation is bounded:

$$|e_i| \leq K_D \, \delta_\infty.$$

Since the clean margin satisfies

$$m_i D_i(x_w) = \Delta_i > K_D \, \delta_\infty$$

we obtain

$$m_i D_i(x_{w'}) = m_i \left( D_i(x_w) + e_i \right) = \Delta_i + m_i e_i \geq \Delta_i - |e_i| > 0$$

Hence, the bit $i$'s sign remains unchanged under overwrite. $\square$

**Theorem 1** (BER upper bound). *For a given triplet $(x, m, m')$ with overwrite $x_{w'}$, the bit error rate satisfies*

$$\operatorname{BER}(x, m, m') \leq \frac{1}{L} \sum_{i=1}^{L} \mathbf{1}\left( \Delta_i(x, m) \leq K_D \delta_\infty \right) + \varepsilon_{\text{rec}} \tag{25}$$

*In particular, if $K_D \delta_\infty < \Delta_{\min}$, then*

$$\operatorname{BER}(x, m, m') \leq \varepsilon_{\text{rec}} \tag{26}$$

*Proof.* Let

$$e_i = D_i(x_{w'}) - D_i(x_w), \qquad \Delta_i = m_i D_i(x_w)$$

A bit $i$ flips if

$$m_i D_i(x_{w'}) \leq 0$$

Substituting gives

$$m_i D_i(x_{w'}) = m_i(D_i(x_w) + e_i) = \Delta_i + m_i e_i$$

Thus, the flip condition is

$$\Delta_i + m_i e_i \leq 0 \quad \implies \quad |e_i| \geq \Delta_i$$

By Lemma 1, each logit deviation is bounded:

$$|e_i| \leq K_D \, \delta_\infty$$

Therefore, a *sufficient* condition for a possible flip is

$$\Delta_i \leq K_D \, \delta_\infty$$

Counting all such bits in the worst case and adding the nominal clean error rate yields the bound. If

$$K_D \, \delta_\infty < \Delta_{\min}$$

then no additional flips can occur beyond nominal errors. □

**Corollary 1** (Local, data-dependent tightening). *Replacing $K_D$ by the local, attack-path constant $K_{D,\text{loc}}(x_w \to x_{w'})$ yields the tighter bound*

$$\text{BER}(x, m, m') \ \leq \ \frac{1}{L} \sum_{i=1}^{L} \mathbf{1}\big(\Delta_i(x, m) \leq K_{D,\text{loc}} \, \delta_\infty\big) \ + \ \varepsilon_{\text{rec}} \tag{27}$$

**Corollary 2** (Perfect recovery under margin condition). *If $\varepsilon_{\text{rec}} = 0$ and*

$$K_D \, \delta_\infty < \Delta_{\min}$$

*then no bits flip under overwrite, and hence*

$$\text{BER}(x, m, m') = 0 \quad \text{for all } (x, m, m')$$

## B.4 EMPIRICAL ESTIMATION OF KEY QUANTITIES

In this subsection, we empirically estimated $\delta_\infty$, $\Delta_{\min}$ and $K_{D,loc}$ to validate the theoretical assumption. All quantities were measured in the $\|\cdot\|_\infty$ norm for consistency with the formal analysis. $K_{D,loc}$ was computed per image as defined in Equation 21. This estimation of the key quantities of our system was done over 3000 sets of images when re-watermarked using our Encoder with the post-processing module. The distributions for $\Delta_{\min}$ and $K_{D,loc}$ are given in Figures 7a and 7b respectively. Table 2 summarizes key statistics for the evaluation of $\Delta_i$, $K_{D,loc}$, $\delta\infty$, and the nominal decoder error over the dataset of 3,000 images. For each metric, the 5th percentile, median, and 95th percentile are reported in Table 2. The median margin $\Delta_{\min} = 3.41$ indicates that the original watermarks are embedded with a strong separation, while the median local Lipschitz constant $K_{D,\text{loc}} = 4.38$ quantifies the typical sensitivity of the decoder to image perturbations. The average bit accuracy is near 100% after overwriting, as shown in Figure 2 across the dataset. This demonstrates that the decoder reliably preserves the original message under self-re-watermarking conditions.

Table 2: Summary statistics (5th percentile, median, 95th percentile) for $\Delta_i$, $K_{D,loc}$, $\delta_\infty$, and Nominal Decoder Error.

| Metric | 5th Percentile | Median | 95th Percentile |
|---|---|---|---|
| $\Delta_{min}$ | 1.8535 | 3.4048 | 4.6087 |
| $K_{D,loc}$ | 3.0627 | 4.3764 | 5.9426 |
| $\delta_\infty$ | 0.9999 | 1.000 | 1.000 |
| $\epsilon_{rec}$ | 0.000 | 0.000 | 0.000 |

## C ADDITIONAL DETAILS ON EXPERIMENTAL SETTINGS

The value $\lambda_{lpips}$ was picked as 0.5, corresponding to empirical evaluation using trained models at various $\lambda_{lpips}$ settings. The results with various $\lambda_{lpips}$ settings are given in Table 3. As per the Table, $\lambda_{lpips}$ as 0.5 gives a good balance of visual fidelity and robustness. Moreover, the adaptive weight adjustment algorithm used during our training is given in Algorithm 2.

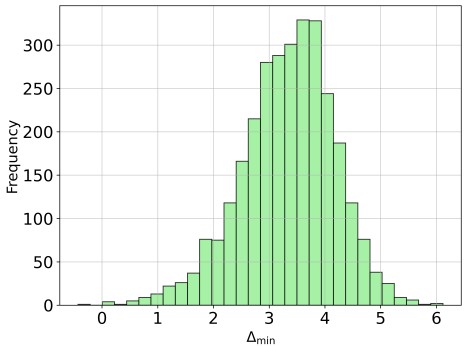 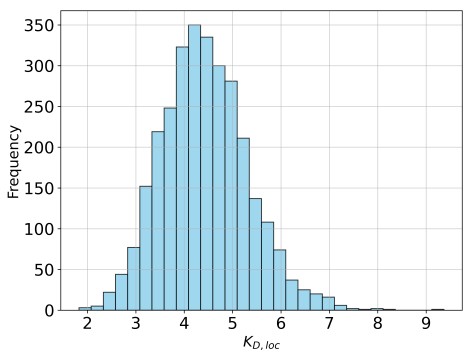

(a) Distribution of $\Delta_{\min}$ over 3,000 images.  (b) Distribution of $K_{D,loc}$ of the decoder.

Figure 7: Comparison of $\Delta_{\min}$ and the empirical Lipschitz constant distributions.

---

**Algorithm 2** Adaptive Weight Adjustment

---

**Require:** Epoch $r$, BER on clean decode $ber\_dec$, BER after overwrite $ber\_over$,
1:    optional previous weights $prev\_w$, smoothing factor $s$, max epochs $R$
**Ensure:** Updated weights $w = \{\lambda_{lpips}, \lambda_{rec}, \lambda_{rob}\}$

2: $dec\_conf \leftarrow 1 - \min\left(\dfrac{ber\_dec}{0.2}, 1\right)$

3: $over\_conf \leftarrow \min\left(\dfrac{ber\_over}{0.2}, 0.5\right)$

4: $trans\_ready \leftarrow \dfrac{dec\_conf + over\_conf}{2}$

5: $epoch\_prog \leftarrow \min\left(\dfrac{r+1}{R}, 1\right)$

6: $\alpha \leftarrow 0.5 \cdot epoch\_prog + 0.5 \cdot trans\_ready$
7: Define $\lambda_{lpips}(\alpha) \leftarrow 4.0 + 5.5 \cdot \alpha$
8: Define $\lambda_{rec}(\alpha) \leftarrow 6.0 - 3.5 \cdot \alpha$
9: Define $\lambda_{rob}(\alpha) \leftarrow 5.0 - 1.0 \cdot \alpha$
10: $target.lpips \leftarrow \lambda_{lpips}(\alpha)$
11: $target.rec \leftarrow \lambda_{rec}(\alpha)$
12: $target.rob \leftarrow \lambda_{rob}(\alpha)$
13: **if** $prev\_w$ is None **then**
14:  $prev\_w \leftarrow target$
15: **end if**
16: **for all** $k \in \{\lambda_{lpips}, \lambda_{rec}, \lambda_{rob}\}$ **do**
17:  $w[k] \leftarrow s \cdot prev\_w[k] + (1-s) \cdot target[k]$
18: **end for**
19: **return** $w$

---

Table 3: LPIPS validation results: Visual Quality and Robustness under different distortions.

| $\lambda_{\text{LPIPS}}$ | Visual Quality | | ACC$_{\text{clean}}$ (%) | | | | | |
|---|---|---|---|---|---|---|---|---|
| | PSNR (dB) | SSIM | JPEG (50) | Gaussian Blur (2.0) | Dropout (30%) | Cropout (30%) | Crop (3.5%) |
| 0.3 | 32.39 | 0.97 | 99.63 | 99.90 | 99.93 | 99.88 | 99.61 |
| 0.5 | 34.03 | 0.97 | 95.06 | 99.66 | 98.90 | 98.14 | 99.85 |
| 0.7 | 39.51 | 0.99 | 88.08 | 99.15 | 99.48 | 99.53 | 92.06 |

## C.1 ADVERSARIAL SETUP

The PGD-based adversarial attack was scheduled using curriculum learning during training. Specifically, the perturbation budget ($\epsilon$) and step size ($\alpha$) are adaptively adjusted as training progresses. Before the start epoch, both values remain near zero to ensure stability; between the start and ramp epochs, $\epsilon$ and $\alpha$ increase linearly to their maximum values; and after the ramp phase, they are fixed at their predefined maxima. The maximum perturbation budget was set to $\epsilon = 0.05$, and the maximum step size was set to $\alpha = 0.009$. The number of iterations was fixed at 50 to balance computational cost with providing the model sufficient exposure to a reasonable attack strength during training. This gradual schedule enables the model to adapt progressively to stronger attacks while maintaining training stability.

## C.2 SOURCE REPOSITORIES OF EVALUATED MODELS

The SOTA models along with the weights evaluated in this work were obtained from the repositories provided by the respective authors and used under their default configurations:

1. HiDDeN: `https://github.com/ando-khachatryan/HiDDeN`
2. MBRS: `https://github.com/jzyustc/MBRS`
3. SSL: `https://github.com/facebookresearch/ssl_watermarking`
4. ARWGAN: `https://github.com/river-huang/ARWGAN`
5. WFORMER: `https://github.com/YuhangZhouCJY/WFormer`
6. VINE: `https://github.com/Shilin-LU/VINE`

# D PERFORMANCE AGAINST ADVERSARIAL SELF-RE-WATERMARKING

## D.1 FURTHER ANALYSIS ON PERTURBATION BUDGET ($\epsilon$) FOR THE PGD ATTACK

We selected the perturbation budget $\epsilon$ for the PGD-based attack such that the attacked image maintains a minimum PSNR of 30 dB relative to the original watermarked image, corresponding to the threshold for acceptable imperceptibility(Subhedar & Mankar, 2020; Zhang et al., 2024). To determine this, we empirically evaluated the PSNR of the watermarked images under attacks with varying values of $\epsilon$. We then identified the values of $\epsilon$ for which at least 90% of the attacked images had PSNR above 30 dB. From this subset, we chose the maximum $\epsilon$ to assess the robustness of our watermarking model against PGD-based attacks. The 10th percentile PSNR as a function of $\epsilon$ is shown in Figure 8a.

Figure 8b shows the bit accuracy under the PGD-based attack for varying values of the perturbation budget $\epsilon$. As $\epsilon$ exceeds 0.05, the model's performance begins to deteriorate and the bit accuracy decreases. Although increasing $\epsilon$ allows the adversary to attempt stronger perturbations, the original watermark remains partially recoverable, and the adversary is unable to fully embed the adversarial watermark.

## D.2 ADVERSARIAL MODEL

We trained two adversarial models on the MIRFLICKR dataset using a learning rate of 0.01. The models differ in their use of the robustness loss:

- **Baseline Adversarial Model (BAM):** Trained with the full algorithm, including the robustness loss term.

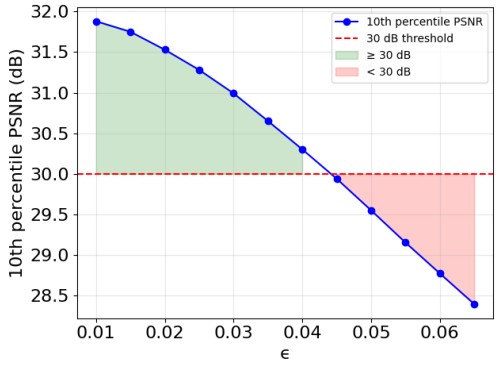
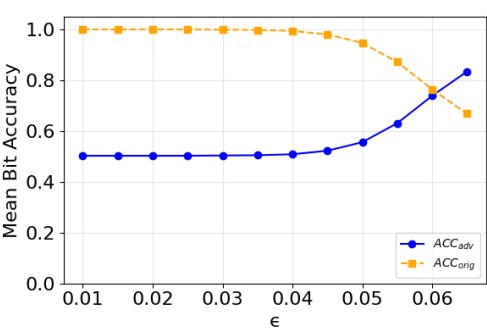

(a) 10th percentile PSNR vs. PGD perturbation budget ($\epsilon$). The dashed line indicates the 30 dB threshold used to select $\epsilon$.

(b) Bit accuracy of original ($\text{ACC}_{\text{orig}}$) and adversarial watermark ($\text{ACC}_{\text{adv}}$) under PGD attack for varying $\epsilon$.

Figure 8: Comparison of PSNR and watermark bit accuracy under varying PGD perturbation budgets.

- **Ablated Model (AM):** Trained using the same algorithm, but with the robustness loss component removed.

Table 4 reports the decoder's ability to recover the original watermark after attacks from each adversarial model.

Table 4: $\text{ACC}_{\text{orig}}$ of the proposed model after attacking with different overwriting models

| Overwriting Model | | $\text{ACC}_{\text{orig}}$ |
|---|---|---|
| Algorithm | Post Processing | |
| BAM | ✔ | 1.00 |
| BAM | ✘ | 1.00 |
| AM | ✔ | 1.00 |
| AM | ✘ | 0.99 |

The performance comparison of these models, including visual quality and the ability to recover *watermark1 (W1)* both before and after self-re-watermarking, is presented in Table 5.

Table 5: The performance of the developed adversarial models in terms of visual quality and message recoverability.

| Adversarial Model | | Visual Quality | | Bit Accuracy | |
|---|---|---|---|---|---|
| Model | Postprocess | PSNR | SSIM | $\text{ACC}_{\text{clean}}$ | $\text{ACC}_{\text{orig}}$ |
| BAM | ✓ | 34.60 | 0.97 | 1.00 | 1.00 |
| BAM | ✗ | 33.00 | 0.97 | 1.00 | 1.00 |
| AM | ✓ | 41.81 | 0.99 | 1.00 | 0.50 |
| AM | ✗ | 38.96 | 0.98 | 1.00 | 0.59 |

# E  MULTI-STAGE RE-WATERMARKING ATTACK

This section extends the discussion beyond direct malicious use of the watermarking models by evaluating an adversary capable of performing multistage re-watermarking attacks. While our work in watermarking focuses on direct malicious re-use of encoders, real-world adversaries may attempt more sophisticated strategies, such as attacking watermarked images with image-processing attacks or sophisticated removal attacks and subsequently re-embedding a new one. Although *multi-stage*

*overwriting*, in which the original watermark is first attempted to be removed and a new one embedded, does not strictly fall under the standard Encoder-Based Self-Re-Watermarking attack, it represents a realistic adversary strategy. In this section, we go beyond our adversarial scope by exploring such multi-stage removal and re-watermarking scenarios. This allows us to assess not only whether watermarks can be overwritten but also whether subtle cues remain that enable ownership verification even under complex adversarial manipulations.

## E.1 ANALYSIS ON IMAGE PROCESSING ATTACKS FOLLOWED BY RE-WATERMARKING

This subsection evaluates the effect of applying multiple image noise operations followed by re-watermarking. We analyze the robustness of our model under a range of image processing attacks, and the results are reported in Table 6. The findings show that our model remains robust under these attacks, followed by self-rewatermarking.

| Image Processing Attack | $ACC_{orig}$ |
|---|---|
| JPEG(80) | 95.09 |
| Gaussian Blur (1.0) | 99.99 |
| Cropout (10%) | 96.07 |
| Dropout (10%) | 99.96 |
| Gaussian Noise (1.0) | 100.00 |
| Histogram Equalization | 99.88 |
| Crop (3.5%) | 99.98 |
| Rotate (10°) | 94.33 |
| Horizontal Flip | 99.53 |
| Vertical Flip | 99.45 |

Table 6: Analysis on various image-processing attacks followed by self-re-watermarking

## E.2 ANALYSIS ON WATERMARK REMOVAL AND RE-WATERMARKING

In this scenario, an adversary first attempts to completely remove the watermark and then re-embeds a new watermark using the same encoder. We empirically evaluated this scenario using CtrlRegen+ (Liu et al., 2025), a state-of-the-art method that demonstrates strong performance for removal attack under both low and high-perturbation watermark settings. The method controls the amount of removal by adjusting the step size. We evaluated our model at step sizes of 0.3, 0.5, and 0.7. The corresponding results are presented in Table 7. The results indicate that overwriting can indeed be successful after the watermark is removed. However, the visual quality of the resulting images is degraded, as evidenced by lower PSNR and SSIM between the watermarked images and the removed-and-re-watermarked images. Figure 9 illustrates that although the semantic information is preserved, the attacked images are blurred and lose color consistency compared with the watermarked images. This is also highlighted by low SSIM scores. Figure 9 shows the watermarked images with the attacked images at different step sizes (0.3, 0.5, 0.7). The removal of the watermark also leaves behind faintly colored artifacts in the center of the image. This suggests that even if the watermark is removed and re-embedded, the original owner can still detect perceptual changes, providing a mechanism to assess whether the content has been altered.

It is important to note that the current study primarily focuses on the direct malicious re-use of encoders for self-re-watermarking attacks. The model's performance under more complex multi-step adversarial scenarios, involving removal followed by re-embedding, is not fully explored. Nevertheless, our preliminary evaluation demonstrates that perceptual degradation in such cases provides an additional cue for ownership verification. Our future work will focus on exploring strategies to mitigate multi-step re-watermarking processes, such as adversarial removal followed by re-embedding of the watermark.

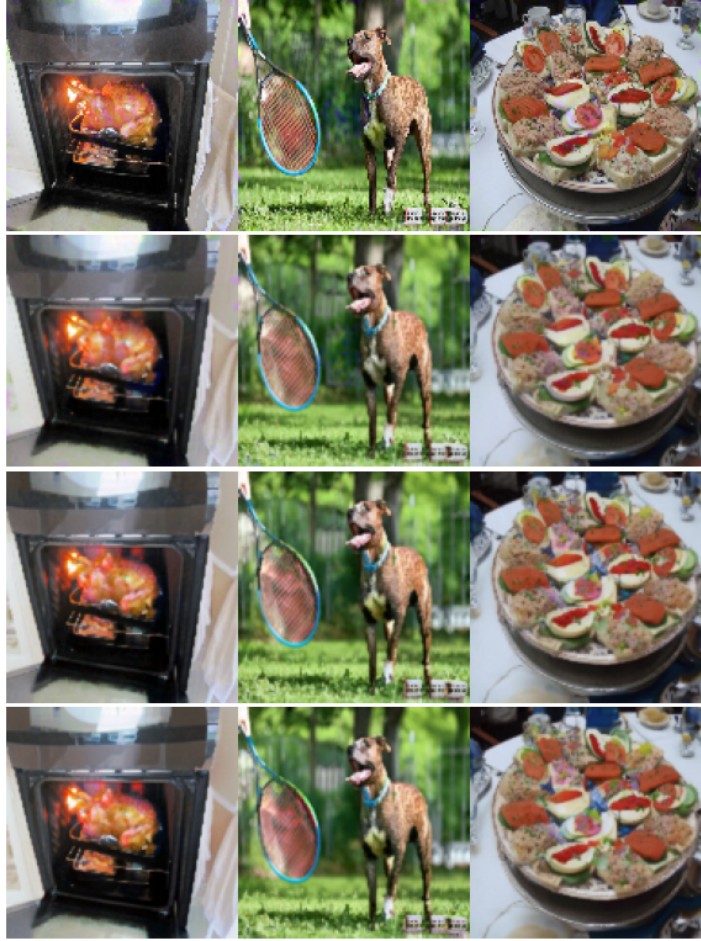

Figure 9: The effect of removal attack on watermarked images. The watermarked image is shown at the top. The attacked images corresponding to step sizes 0.3, 0.5, and 0.7 are displayed in rows 2, 3, and 4, respectively. Please zoom in for a closer look.

Table 7: Performance metrics at different steps of removal attack

| Step | $ACC_{adv}(\%)$ | $ACC_{orig}(\%)$ | PSNR | SSIM |
|------|------|------|------|------|
| 0.3 | 78.34 | 68.11 | 23.27 | 0.72 |
| 0.5 | 87.41 | 56.97 | 21.85 | 0.65 |
| 0.7 | 90.83 | 53.00 | 20.50 | 0.59 |

# F ADDITIONAL EXPERIMENTS

## F.1 EVALUATION OF ROBUSTNESS AGAINST IMAGE PROCESSING ATTACKS

To complement this quantitative analysis, Figure 10 presents box plots of bit accuracy distributions under pixel-wise distortions (e.g., Gaussian noise, salt-and-pepper noise), while Figure 11 illustrates performance under geometric attacks (e.g., StirMark-style elastic deformation (Petitcolas et al., 1998), rotation, flipping, and scaling). The proposed model consistently shows low variance across the test images, underscoring its stability under real-world perturbations.

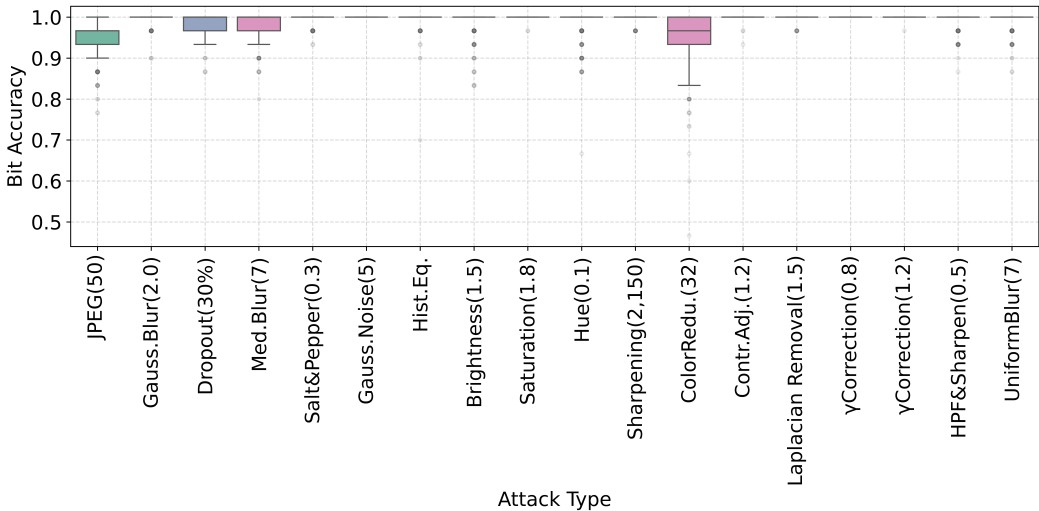

Figure 10: Bit accuracy distribution under pixel-wise distortions.

To further contextualize the robustness of the proposed model, we compared its performance against a baseline model that uses the same architecture but without spectral normalization. As shown in Figure 12, the proposed method consistently outperforms the baseline across varying intensities of JPEG compression, Gaussian blur, dropout, and cropping. The baseline exhibits a marked decline in accuracy as distortion severity increases, particularly under aggressive cropping and dropout. In contrast, the spectrally normalized model maintains stable performance under these conditions. This comparison reinforces the practical effectiveness of the proposed design in maintaining watermark integrity under diverse and challenging conditions. Further, Figures 10 and 11, demonstrate that the proposed model achieves higher median bit accuracy and exhibits significantly lower variance under different pixelwise and geometric perturbations.

Furthermore, the proposed model is evaluated on additional benchmark datasets, CelebA (Liu et al., 2015), MIRFLICKR-1M (Huiskes & Lew, 2008), and ImageNet (Deng et al., 2009), to assess its generalizability. For each dataset, 3,000 images were randomly sampled. The visual quality of the outputs and the model's robustness against commonly studied image processing attacks are summarized in Table 8. The proposed model achieves consistently high visual quality across multiple datasets, with PSNR around 33–34 dB and SSIM near 0.96–0.97. It also demonstrates strong robustness to various distortions, maintaining over 94% accuracy under JPEG compression and over 99% under Gaussian blur, dropout, cropout, and small cropping, highlighting its generalizability across diverse image distributions.

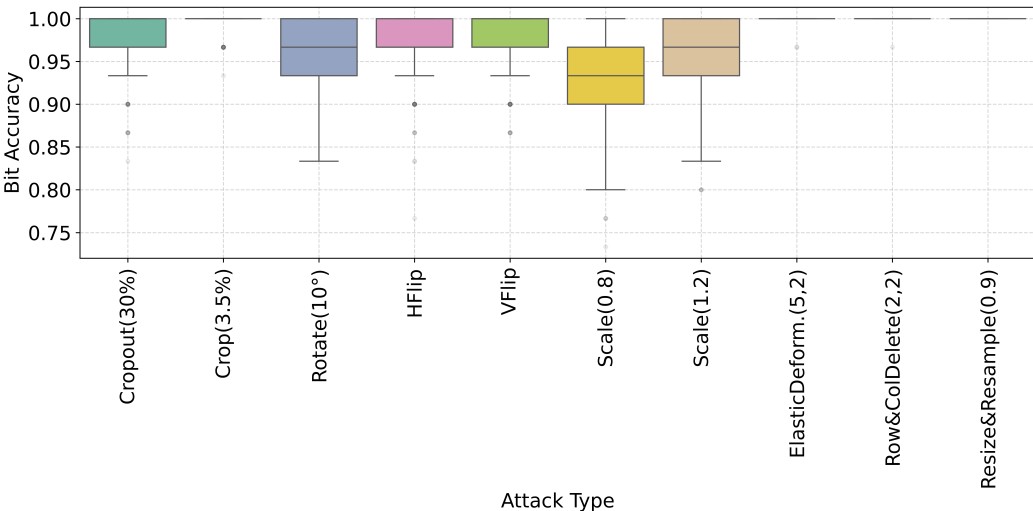

Figure 11: Bit accuracy distribution under geometric distortions.

Table 8: Visual Quality and Robustness (%) across Datasets.

| Studies | Visual Quality | | ACC$_{\text{clean}}$ (%) | | | | | ACC$_{\text{orig}}$ (%) | | |
|---|---|---|---|---|---|---|---|---|---|---|
| | PSNR (dB) | SSIM | JPEG (50) | Gaussian Blur (2.0) | Dropout (30%) | Cropout (30%) | Crop (3.5%) | Self Re-embed | PGD Moderate | PGD Strong |
| COCO | 34.03 | 0.97 | 95.06 | 99.66 | 98.90 | 98.14 | 99.85 | 100.00 | 99.95 | 99.37 |
| MIRFLICKR | 33.48 | 0.96 | 94.46 | 99.48 | 98.81 | 97.75 | 99.68 | 100.00 | 99.76 | 98.85 |
| CELEBA | 34.55 | 0.96 | 95.17 | 99.74 | 98.57 | 97.95 | 99.91 | 99.87 | 99.23 | 99.10 |
| ImageNet | 33.65 | 0.97 | 94.65 | 99.58 | 98.71 | 97.73 | 99.80 | 100.00 | 99.81 | 99.09 |

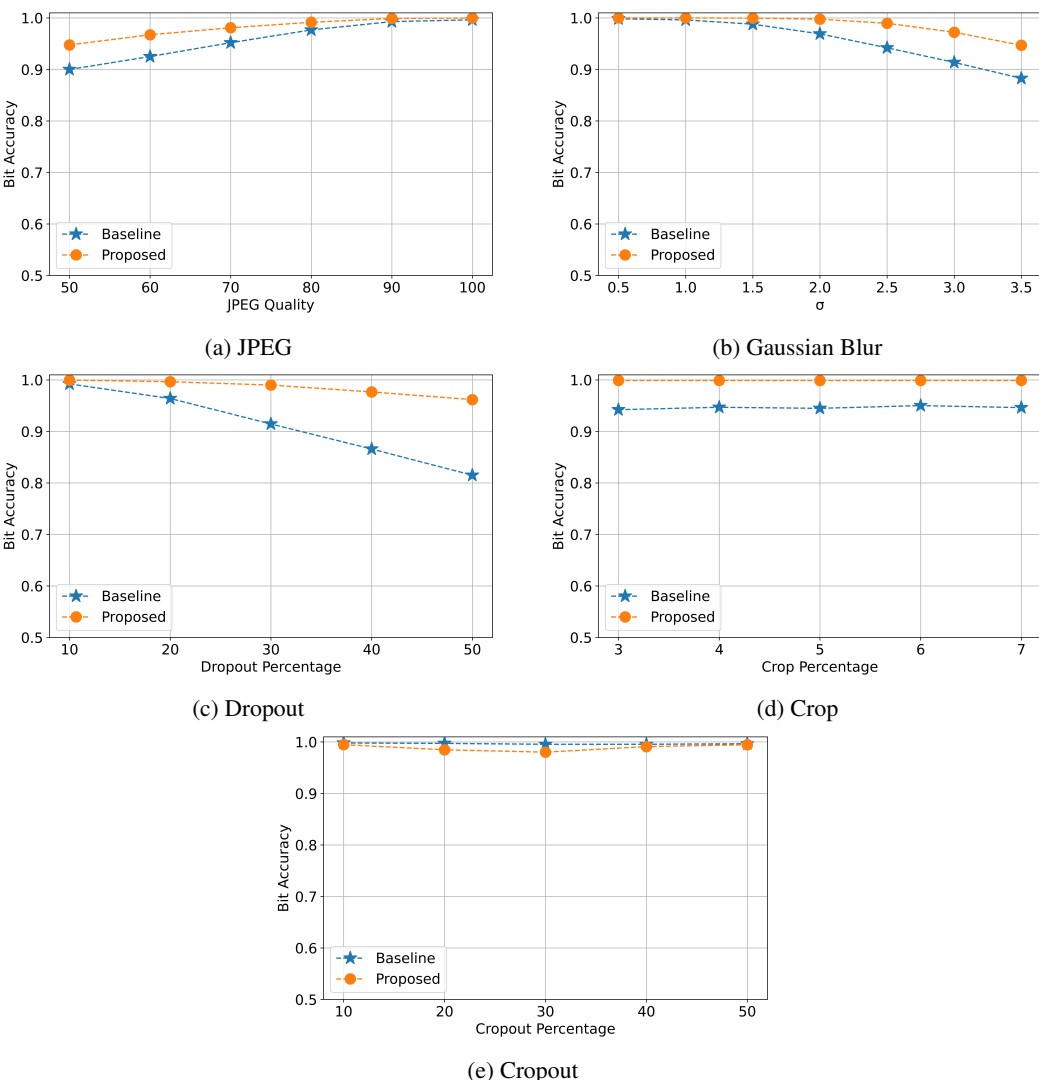

Figure 12: Robustness evaluation under different distortions. (a) JPEG compression. (b) Gaussian blur. (c) Dropout. (d) Crop. (e) Cropout.

## F.2    RESOURCE UTILIZATION

Table 9 presents a comparison of FLOPs and trainable parameters across several deep learning-based watermarking models. Despite the high parameter count (37.09M) from fully connected (FC) layers used for message up- and downsampling, the model's computational cost remains low (7.73 GFLOPs), since most FLOPs are incurred by convolutions over spatial feature maps rather than the low-dimensional FC operations. Although training may take longer, the model provides fast inference and strong watermarking performance. Experimental analysis shows that encoding is achieved at 43.97 images per second, while decoding reaches 607.54 images per second under normal load conditions.

Table 9: Resource Utilization

| MODEL | FLOPS (G) | Trainable Parameters (M) |
|---|---|---|
| HiDDeN | 6.72 | 0.41 |
| MBRS | 13.35 | 5.80 |
| ARWGAN | 24.22 | 2.30 |
| WFORMER | 14.83 | 1.72 |
| OURS | 7.73 | 37.09 |

## F.3    SCALABILITY TO HIGHER RESOLUTIONS

We further evaluated the proposed architecture using higher image resolution and payload size, setting the image dimensions to $256 \times 256$ and the payload to 64 bits. The empirical evaluation shows that the model's performance remains consistent with that of the $128 \times 128$, 30-bit configuration, thereby confirming the scalability of the proposed approach. The average visual quality was assessed using PSNR (32.72 dB) and SSIM (0.98). In addition, the model demonstrates the ability to withstand self-overwriting when the same encoder is used, ensuring reliable recovery of the original message even if it is overwritten. Furthermore, analysis under PGD attacks revealed that 99.94% of the original message could be recovered at the moderate level, and 99.86% at the strong level. A detailed robustness evaluation against common image processing attacks is provided in Table 10.

Table 10: Robustness performance comparison

| JPEG (50) | Gaussian Blur (2.0) | Dropout (30%) | Cropout (30%) | Crop (3.5%) |
|---|---|---|---|---|
| 99.06 | 99.78 | 96.28 | 98.24 | 97.63 |

## F.4    CROSS MODEL RE-WATERMARKING

In this subsection, we analyze the robustness of our system to withstand cross-model overwriting. Table 11 reports $\text{Acc}_{\text{orig}}$ after the images are encoded by different watermarking models. The results indicate that the original watermark can be effectively recovered by our model even under cross-model overwriting.

| Model | $\text{ACC}_{\text{orig}}$ |
|---|---|
| SSL | 100.00 |
| WFORMER | 99.35 |
| HiDDeN | 99.23 |
| MBRS | 99.52 |
| ARWGAN | 99.23 |
| VINE | 100.00 |

Table 11: Robustness of our model to cross-model overwrites

## F.5 ABLATION STUDY

In this subsection, we evaluate the impact of the post-processing module as well as the contribution of spectral normalization in the proposed system. We assess model performance both in terms of visual quality and the decoder's ability to recover the embedded watermark under benign and attacked scenarios. The quantitative results are summarized in Table 12. The model trained without spectral normalization can successfully recover the original watermark after an adversarial attack, but fails to do so under self-re-watermarking. This further confirms that spectral constraints enhance robustness and validate our design choices.

Table 12: Ablation Study Results

| Model | Visual Quality | | $ACC_{clean}$ | $ACC_{orig}$ | | |
| --- | --- | --- | --- | --- | --- | --- |
| | PSNR | SSIM | | After Self OW | After moderate PGD | After strong PGD |
| Proposed | 34.03 | 0.97 | 100.00 | 100.0 | 99.95 | 99.37 |
| Proposed w/o Post Processing | 31.82 | 0.96 | 100.00 | 100.00 | 100.00 | 99.99 |
| Proposed w/o Spectral Norm | 30.40 | 0.94 | 99.90 | 76.33 | 99.57 | 98.90 |

Table 12 illustrates that although the post-processing module reduces message recoverability by a very small margin, it provides a boost in visual quality. This suggests that the decision to use the auxiliary post-processing module depends on the scenario in which the watermarked images are used. It can be enabled for fidelity focused applications where visual quality is paramount and can be disabled for security focused applications where robustness and forensic recoverability are required.

## G ANALYSIS ON THE BEHAVIOR OF THE SYSTEM

In this section, we analyze the distortion that occurs when the Encoder attempts to apply a watermark to an image that has already been watermarked. Fast Fourier Transform (FFT) analysis reveals that such re-watermarking introduces significant high-frequency artifacts, indicating substantial distortion. Similarly, pixel intensity histograms show a shift in the distribution toward brighter regions, reflecting altered image characteristics. This behavior is a direct consequence of the training pipeline, which uses asymmetric optimization where the Encoder is guided by a Fidelity Loss to ensure the initial watermark remains imperceptible, while the visual quality of the re-watermarked image is not optimized. Instead, the re-watermarked image is treated as an adversarial case, with only the Decoder being optimized via a Robustness Loss to successfully recover the original message from a heavily distorted image.

The images presented in Figures 13 and 14 serve as illustrative examples of this phenomenon.

## H EXTENDED RELATED WORK

To contextualize our study, this section reviews two core areas: advances in deep learning–based watermarking and the evolving adversarial threats and countermeasures.

### H.1 DL BASED IMAGE WATERMARKING

Deep learning has become central to image watermarking, enabling models to learn optimal trade-offs between imperceptibility and robustness. Early approaches, such as Baluja et al. (Baluja, 2017), demonstrated the feasibility of steganography using DL, while HiDDeN (Zhu et al., 2018) introduced differentiable noise layers during training to simulate distortions like cropping, compression, and blurring. To address non-differentiable or unknown distortions, Luo et al. (Luo et al., 2020) proposed a distortion-agnostic framework using adversarial training and channel coding. MBRS (Jia et al., 2021) further improved robustness to JPEG compression by incorporating both real and simulated codecs into the training loop. Other advances, such as ARWGAN (Huang et al., 2023) employed attention-based feature fusion to improve robustness, though often at high computational cost. Fernandez et al. (Fernandez et al., 2022) applied self-supervised learning with DINO (Caron et al., 2021) to embed watermarks in semantically meaningful regions, improving removal and synchronization resistance, but being vulnerable to cropping.

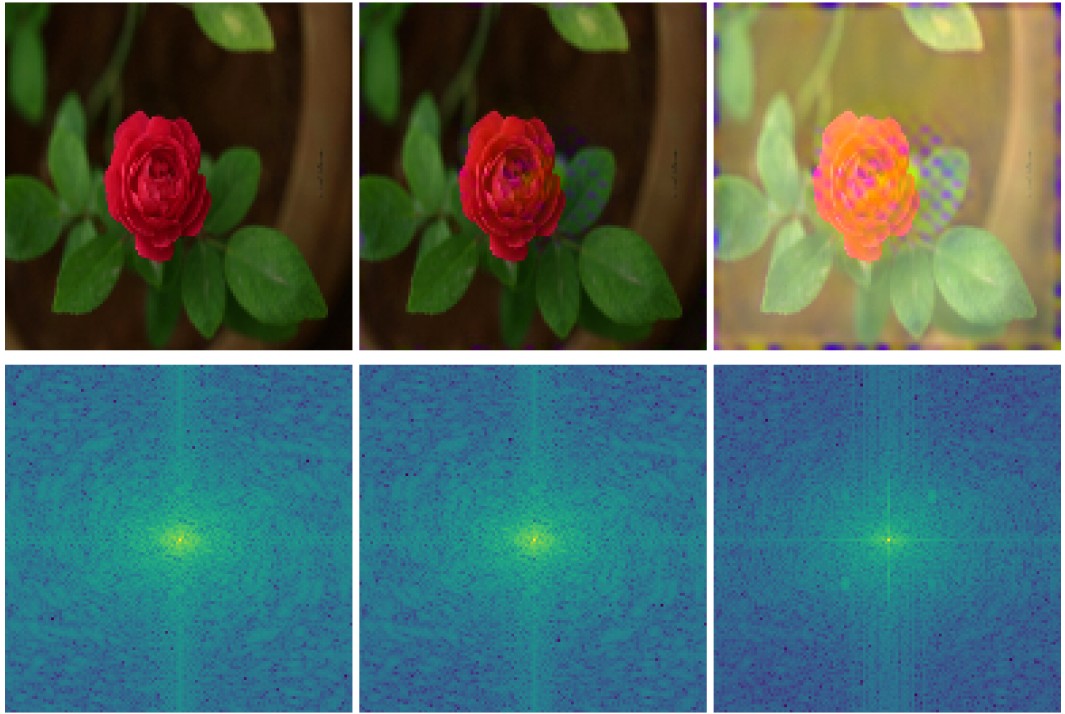

Figure 13: Comparison of the frequency components of the cover, watermarked, and re-watermarked images.

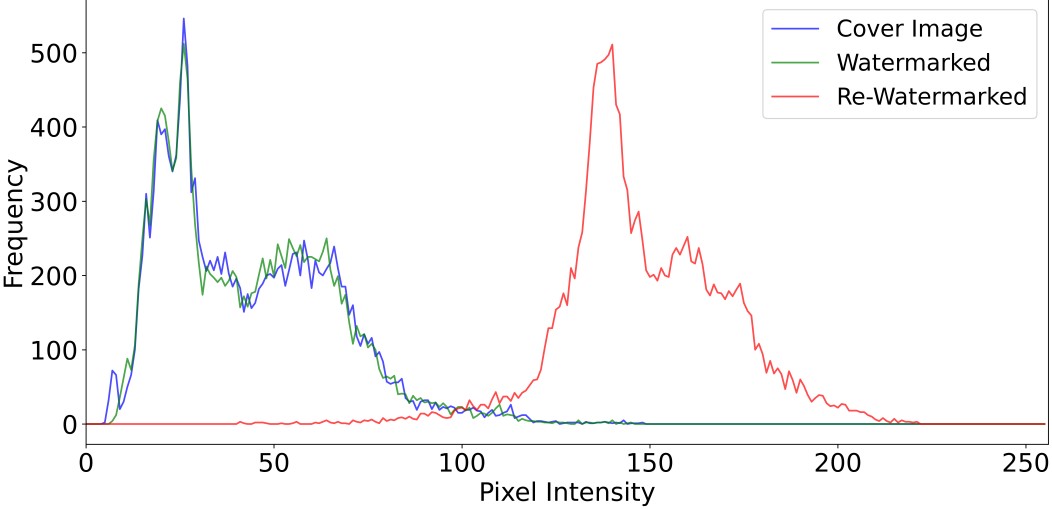

Figure 14: Histogram illustrating the distribution of pixels in the cover, watermarked, and re-watermarked images. All three exhibit a similar overall pattern, although the re-watermarked image shows a noticeable shift toward brighter pixel values.

WFormer (Luo et al., 2024) leveraged Transformer-based encoding and soft fusion to improve robustness and imperceptibility across standard distortions, but did not address adversarial or security-focused threats. Although VINE (Lu et al., 2025) developed a robust model against image editing, the systematic vulnerability of self-re-watermarking still remains. GANMarked (Singh et al., 2024) tackled security via key-based protection layers, offering some protection against unauthorized extraction, but showing limited resilience to compression and forgery. Some recent dual watermarking methods (Padhi et al., 2024b) and adversarially trained visible watermarks attempt to counter model style-transfer attacks but lack robustness under encoder reuse. In contrast, VINE(Lu et al., 2025) focuses on addressing the specific vulnerabilities introduced by large-scale text-to-image models by utilizing a powerful generative prior and frequency-based surrogate attacks to embed watermarks that are resistant to common image editing techniques.

Despite recent progress, a key vulnerability remains underexplored: *self-re-watermarking*, where the same encoder is maliciously reused to embed a new message into a watermarked image. Most existing systems lack mechanisms to detect or resist such attacks due to open encoding pipelines. This highlights the need to shift focus from decoder-side defenses to encoder-level robustness against overwriting.

## H.2 ADVERSARIAL ATTACKS IN DL BASED IMAGE WATERMARKING

Deep learning-based image watermarking systems face a range of adversarial threats that aim to compromise their security guarantees. A significant yet underexplored risk is *self-re-watermarking*, where an adversary reuses the encoder to embed a new message into an already watermarked image. Unlike removal attacks (Zhao et al., 2024; An et al., 2024), which attempt to erase the embedded watermark and thus invalidate ownership, re-watermarking introduces a conflicting ownership claim, fundamentally undermining the reliability of watermark-based provenance.

Kinakh et al. (2024) highlighted related risks by demonstrating that self-supervised watermarking techniques are prone to unauthorized transfer, suggesting the availability of model-related information itself as a potent attack vector. Further, existing literature has documented other adversarial vectors against watermarking systems. Forgery-based threats (Hu et al., 2025) generate counterfeit ownership claims. These studies collectively underscore that adversarial pressure on watermarking systems is expanding in scope and sophistication.

Defensive strategies have been proposed to mitigate watermarking threats. For example, diffusion-based approaches (Zhu et al., 2024) introduce adversarial examples containing personalized watermarks to obstruct unauthorized imitation by generative models. In addition, frameworks such as Watermark Vaccine (Liu et al., 2022) and Universal Watermark Vaccine (Chen et al., 2023) leverage adversarial learning to immunize models against the removal of visible watermarks. However, while much of this work focuses on defenses against removal attacks of visible watermarks, relatively little attention has been paid to defenses against overwriting attacks. Among these, Chen et al. (2024b) designed a scheme resistant to model-based overwriting, but its generalization beyond that scenario is limited. Padhi et al. (2024a) proposed a dual-watermarking method that provides robustness against surrogate overwriting attacks. Despite these efforts, robust countermeasures to self-re-watermarking remain largely absent.

Building on this analysis, we identify self-re-watermarking as a critical yet overlooked vulnerability in existing watermarking systems. To directly address this gap, we propose a proactive framework that limits the model's sensitivity to input changes. This helps the developed models to defend against self-overwriting while preserving robustness against conventional image processing attacks. In doing so, our approach broadens the scope of watermarking defense beyond removal-centric strategies and establishes resilience against the emerging threat of adversarial re-watermarking.

# I  ADDITIONAL QUALITATIVE RESULTS

This section presents visual examples of the original watermarked images alongside their re-watermarked versions. These comparisons illustrate how the proposed method preserves the embedded message and maintains image quality even under self-overwriting attacks.

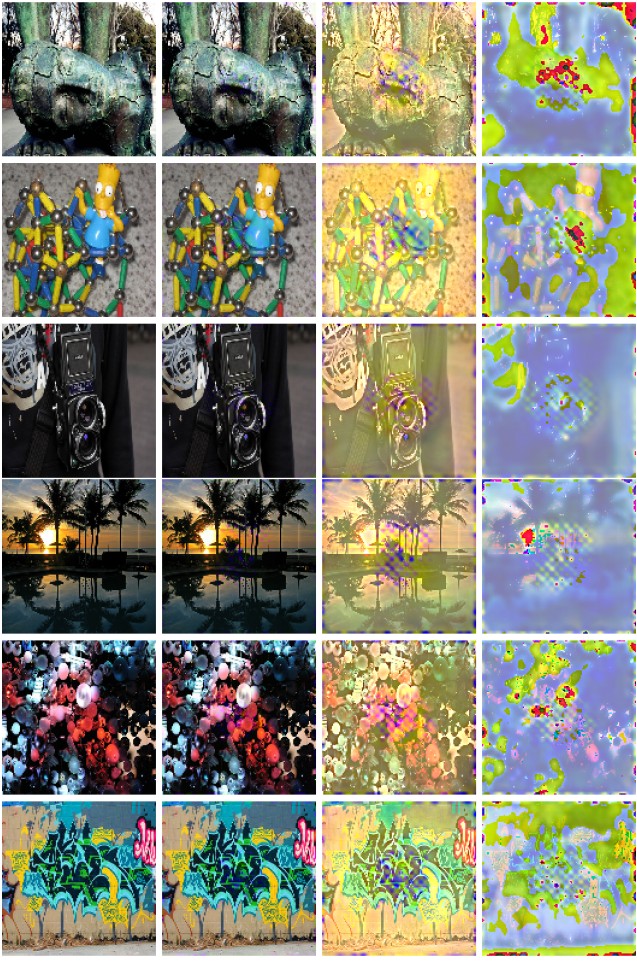

Figure 15: Qualitative Results: First column shows the original images, second column shows the watermarked images, third column shows the re-watermarked images, and the fourth column shows the difference between watermarked and re-watermarked images.

## LARGE LANGUAGE MODEL USAGE

Large language models were used solely to lightly polish the writing and improve grammar; they were not used for generating ideas.

