# OpenReview forum: "THE SELF-RE-WATERMARKING TRAP: FROM EXPLOIT TO RESILIENCE"
_ICLR.cc/2026/Conference — ICLR 2026 Poster_

### Official Review · Reviewer_bosH · 2025-10-21

**Soundness:** 4
**Presentation:** 3
**Contribution:** 4
**Rating:** 8
**Confidence:** 3

**Summary:**

This paper introduces the self-re-watermarking threat model, where an attacker can reuse the watermark encoder to embed a second watermark into an already watermarked image. The authors demonstrate that existing Encoder-Decoder-based watermarking methods are not robust against this type of attack and often recover the second watermark instead of the original one. To solve this problem, the paper presents a methodology that combines Lipschitz continuity and adversarial training to limit the effect of re-watermarking attacks. They evaluate this approach against both a simple self-re-watermarking attack and a gradient-based re-watermarking attack.

**Strengths:**

1. Novel threat model and contributions. It is akin to performing both a removal attack on the existing watermark and a forgery to add a new watermark.
2. Empirically and theoretically motivated.
3. Good results.

**Weaknesses:**

1. I did not find major weaknesses with the paper; using a second dataset for the experiments to show that the phenomenon is not dataset-dependent could be valuable.
2. The notation should be included in the main paper rather than the appendix. The paper should be readable without the appendix, which is not the case without the notation.
3. The writing could be improved. There is some redundancy in the writing that, if removed, could free enough space to include the notation in the paper. I provide more detail in the suggestions.

**Questions:**

Questions:
1. I am struggling to rationalize how, with your method, the decoder is sensitive enough to embed a watermark, but not sensitive enough to re-watermark an already watermarked image. Did you look at the delta-K plot when comparing x and x_w? For this to work, it would need most points below the y = x line. However, if so, what makes watermarking the first time and re-watermarking so intrinsically different?
2. In Section 5.4, you claim to evaluate the worst-case scenario, but use the local K bound rather than the global bound. Did you look at the global bound, and if so, did you find any meaningful difference?
3. Did you study re-watermarking with a different watermarking scheme than the original watermark?
4. This last question is both a question and a suggestion. Did you study how your methodology affects the performance of removal and forgery attacks? Intuitively, your GBA attack behaves similarly to an adaptive removal attack if one sets the target to be random bits, similar to [1].

Suggestions:
1. The training objective and pipeline could be combined and shortened, with the lengthy description of the loss replaced with an equation. Also, Section 4.2 does not convey any meaningful information beyond what was already said earlier and could be removed.
2. I think having Sections 3.3 and 3.4 come before Sections 3.1 and 3.2 would improve readability and clarity, making the paper feel less redundant.

Typos:
1. Line 89, no colon after follows and a trailing period after the bullet point.
2. The word “based” in line 252 does not make sense to me.
3. Line 442-443, there is a space missing after the period at the end of the sentence.

[1] Lukas, Nils, et al. "Leveraging optimization for adaptive attacks on image watermarks." arXiv preprint arXiv:2309.16952 (2023).

---

> ### Author Response · Authors · 2025-11-20
> **Responses to Review by Reviewer bosH**
>
> We thank the reviewer for their encouragement and constructive feedback. We have addressed all your concerns point by point below.
>
> ---
>
> **Weakness 1: I did not find major weaknesses with the paper; using a second dataset for the experiments to show that the phenomenon is not dataset-dependent could be valuable.**
>
> Following the suggestion, we have included the results of our model’s robustness on three additional datasets, namely MIRFLICKR, CelebA, and Imagenet. The analysis includes robustness to various image processing attacks, self-re-watermarking attacks, and the gradient-based attack. The results have been presented in Appendix F.1(page 24). The performance is consistent with the results observed on the COCO dataset.  For your convenience, we have included the table below. Kindly note that the last eight columns report the bit accuracy of the original watermark after each respective attack.
> | Datasets     | PSNR (dB) | SSIM | JPEG (50) | Gaussian Blur (2.0) | Dropout (30%) | Cropout (30%) | Crop (3.5%) | Self Re-embed | PGD Moderate | PGD Strong |
> |------------|------------|------|-----------|--------------------|---------------|---------------|-------------|---------------|--------------|------------|
> | COCO       | 34.03      | 0.97 | 95.06     | 99.66              | 98.90         | 98.14         | 99.85       | 100.00        | 99.95        | 99.37      |
> | MIRFLICKR  | 33.48      | 0.96 | 94.46     | 99.48              | 98.81         | 97.75         | 99.68       | 100.00        | 99.76        | 98.85      |
> | CELEBA     | 34.55      | 0.96 | 95.17     | 99.74              | 98.57         | 97.95         | 99.91       | 99.87         | 99.23        | 99.10      |
> | ImageNet   | 33.65      | 0.97 | 94.65     | 99.58              | 98.71         | 97.73         | 99.80       | 100.00        | 99.81        | 99.09      |
>
>
> >**Weakness 2: The notation should be included in the main paper rather than the appendix. The paper should be readable without the appendix, which is not the case without the notation.**
>
> We have revised our manuscript to include all the notations in the main manuscript.
>
>
> >**Weakness 3: The writing could be improved. There is some redundancy in the writing that, if removed, could free enough space to include the notation in the paper. I provide more detail in the suggestion**
>
> >**Suggestions:**
>
> >**1.The training objective and pipeline could be combined and shortened, with the lengthy description of the loss replaced with an equation. Also, Section 4.2 does not convey any meaningful information beyond what was already said earlier and could be removed.**
>
> >**2. I think having Sections 3.3 and 3.4 come before Sections 3.1 and 3.2 would improve readability and clarity, making the paper feel less redundant.**
>
> We have carefully considered the feedback and have revised the manuscript accordingly. Specifically, we have shortened the training objective and pipeline sections by replacing the description of the loss with equations.  We have also removed Section 4.2. We have revised Section 3 so that Sections 3.3 and 3.4 now precede Sections 3.1 and 3.2.

---

> > ### Author Response · Authors · 2025-11-20
> >
> > > **Question 1: I am struggling to rationalize how, with your method, the decoder is sensitive enough to embed a watermark, but not sensitive enough to re-watermark an already watermarked image. Did you look at the delta-K plot when comparing x and x_w? For this to work, it would need most points below the y = x line. However, if so, what makes watermarking the first time and re-watermarking so intrinsically different?**
> >
> >
> > Firstly, we would like to clarify the delta-K plot in Figure 2 (Figure 4 in the revised manuscript). The plot in Figure 2 (Figure 4 in the revised manuscript) empirically evaluates how the bound we derived in our formal analysis holds in practice. It shows whether the most vulnerable bits of the watermark for each image in our dataset flip under overwriting. In theory, points in the delta-K plot need to lie above the $y = x$ line for a bit flip to be avoided. Points below the line indicate potential vulnerability. Empirically, however, due to the strong robustness of our decoder, most bits remain unflipped under overwriting scenarios, even when lying below the line. Further, this validates our Proposition 1, that if $\Delta_{i} > K_{D,loc}\delta$, then bit  $i$ flip cannot occur.
> >
> > We examined watermarked and re-watermarked images during the plotting of the delta-K plot, but looking at the images alone did not lead to a clear conclusion. To understand the model’s behavior, we analyzed re-watermarking using Fast Fourier Transform (FFT) and histogram analyses. These analyses reveal that the encoder pushes pixels toward higher-intensity regions during re-watermarking, whereas during the original watermarking process, it preserves pixel relationships close to the cover image.
> >
> > This behavior reflects our asymmetric training design. The Encoder is optimized via a Fidelity Loss to ensure the initial watermark remains imperceptible. However, for the re-watermarking scenario, the visual quality is not optimized. Consequently, when the Encoder attempts to embed a second watermark over the existing signal, it is not penalized for distortion. During re-watermarking, the structured distortions in the watermarked image force the Encoder to push pixels into high-intensity regions, creating the statistical artifacts observed in our FFT and Histogram analysis. Since the Decoder is trained to be robust against precisely this type of high-distortion noise, it effectively filters out these artifacts and recovers the original watermark. These analyses and discussions are included in Appendix G (page 27) of the revised manuscript.
> >
> > >**Question 2: In Section 5.4, you claim to evaluate the worst-case scenario, but use the local K bound rather than the global bound. Did you look at the global bound, and if so, did you find any meaningful difference?**
> >
> > In Section 5.4, we evaluated the worst-case scenario by analyzing whether the most vulnerable bit watermarked in each image of our test dataset flips under overwriting. The most vulnerable bit corresponds to the bit with the lowest $\Delta_i$. We used the local $K_{D}$ value for each image, as it provides clearer insight into the behavior of flipping for each tested image rather than across the entire dataset. Regarding the difference between the global and local $K_D$ bounds, we estimated the value of global $K_D$ by sampling small noise perturbations $\ell_\infty$ around the watermarked images. This produced an empirical global $K_D$ value of 38.6 on our test dataset. The global $K_D$ value is larger than  95% of $K_{D,loc}$ values because it measures the sensitivity of the decoder in every possible perturbation direction rather than in the direction of the overwrite. Therefore, using the global $K_D$ yields a conservative bound. In contrast, using the local $K_D$ evaluated specifically on the overwrite path produces a much tighter and more informative per-bit bound for the bound model we studied in Section 5.4.

---

> ### Author Response · Authors · 2025-11-20
>
> >**Question 3: Did you study re-watermarking with a different watermarking scheme than the original watermark?**
>
> Yes, we investigated the effect of re-watermarking with a different watermarking scheme than the original one, which is called cross-model overwriting, for our model with several baselines. For each baseline, including SSL, MBRS, ARWGAN, HIDDEN, WFORMER, and VINE, we independently re-watermarked the images produced by our model and then evaluated recovery. In all cases, the original watermark message remained recoverable by our decoder with accuracy greater than 99%, indicating that our scheme is robust under cross-model overwriting. We have included the results in Appendix F.4 (page 26) of the revised manuscript. We have included the results table below.
>
> | **Model** | **ACC$_{\text{orig}}$** |
> |-----------|------------------------|
> | SSL       | 100.00                 |
> | WFORMER   | 99.35                  |
> | HiDDeN    | 99.23                  |
> | MBRS      | 99.52                  |
> | ARWGAN    | 99.23                  |
> | VINE      | 100.00                 |
>
> >**Question 4: This last question is both a question and a suggestion. Did you study how your methodology affects the performance of removal and forgery attacks? Intuitively, your GBA attack behaves similarly to an adaptive removal attack if one sets the target to be random bits, similar to [1].**
>
> We have studied the effect of removal and forgery attacks. Specifically, we studied watermark removal as proposed in [1], which introduces two types of removal attacks, namely Adversarial Noising and Adversarial Compression. As the reviewer correctly noted, the adversarial noising attack is similar to ours, but the key difference is that its objective explicitly aims to remove the watermark. Our results indicate that our model is robust to the Adversarial Noising attack, likely due to the adversarial training setup used in our method.
>
> However, our model is vulnerable to the Adversarial Compression attack, but, the attack results in noticeable degradation of visual quality (PSNR 19.25 dB, SSIM 0.59). In addition, we evaluated the effect of another advanced removal algorithm, CtrlGen+ [2]. These experiments show that watermark removal is possible, but again, the resulting images have lower visual quality (PSNR < 24dB, SSIM < 0.73). The results for CtrlGen+ have been included in Appendix E.2 (page 22).
>
> In our study, we also examined forgery attacks, specifically cross-model re-watermarking and self-re-watermarking. Extensive experiments demonstrate that our model is robust against these attacks. We have included the results about Cross Model re-watermarking in Appendix F.4 (page 26) and self-re-watermarking in Section 5.3 (page 8). Other types of forgery attacks are beyond the scope of this work, and our future work will focus on defending against them.
>
> [1] Lukas, Nils, et al. "Leveraging optimization for adaptive attacks on image watermarks." arXiv preprint arXiv:2309.16952 (2023)
>
> [2] Yepeng Liu, Yiren Song, Hai Ci, Yu Zhang, Haofan Wang, Mike Zheng Shou, and Yuheng Bu. Image watermarks are removable using controllable regeneration from clean noise. In The Thirteenth International Conference on Learning Representations, 2025. URL https: //openreview.net/forum?id=mDKxlfraAn
>
>
> > **Typos:**
>
> > **1. Line 89, no colon after follows and a trailing period after the bullet point.**
>
> > **2. The word “based” in line 252 does not make sense to me.**
>
> > **3. Line 442-443, there is a space missing after the period at the end of the sentence.**
>
> We have carefully proofread the manuscript to correct grammatical errors and typos, including the above three typos identified by the reviewer.

---

> ### Author Response · Authors · 2025-11-24
>
> Dear Reviewer,
>
> Thank you once again for your thoughtful comments and valuable feedback.
>
> We have made every effort to address all of your suggestions and concerns, and we would be grateful if you could review our rebuttal and let us know if any questions or concerns remain.
>
> Thank you very much for your time and consideration.
>
> Best regards,
>
> Authors

---

> > ### Comment · Reviewer_bosH · 2025-11-25
> > **Response**
> >
> > Thank you for your detailed rebuttal and for implementing the desired changes. I don't have any more questions or concerns.

---

> ### Author Response · Authors · 2025-11-26
>
> Dear Reviewer,
>
> We are happy to hear that our rebuttal addressed all your concerns.
>
> Thank you very much.
>
> Best regards,
>
> Authors

---

### Official Review · Reviewer_QCmH · 2025-10-27

**Soundness:** 2
**Presentation:** 3
**Contribution:** 2
**Rating:** 4
**Confidence:** 4

**Summary:**

The paper introduces a threat model called self-re-watermarking, where an attacker reuses the same encoder that embeds a new message into an already watermarked image. This can hijack ownership by making the decoder recover the attacker’s message.
Experiments demonstrate that existing deep watermarking systems fail under such an attack. They propose a defense solution with Lipschitz constraints to preserve the original watermark.

**Strengths:**

1. The paper is well-structured, with a clear progression.
2. They apply Lipschitz constraints to watermarking and directly training against self-overwriting is reasonable and practically motivated.
Using a composite loss that explicitly accounts for overwrite scenarios is aligned with the threat model.
3. The paper compares against several well-known watermarking methods (HiDDeN, MBRS, etc).

**Weaknesses:**

1. Cross-model overwriting is more frequent than self-re watermarking for attackers to do in many real-world settings. Protectors , leaking the encoder is not accepted in practical applications. Protectors usually do not use the encoder after it leaks. The motivation or task setting is not reasonable.
2. The defense mechanisms (Lipschitz awareness) are known techniques for bounding sensitivity; the novelty lies more in applying them to watermarking. This is fine but it means the new contribution is primarily the threat framing and integration, not a fundamentally new robustness technique.
3. All experiments use 128×128 images and 30-bit payloads. Many watermarking applications use higher resolutions (e.g., 256–1024) and variable payload sizes (32–100+ bits).
4. The boxplot can be refined for better visibility, such as setting different intervals for each sub-figure in Fig. 1.

**Questions:**

Can you add the framework and visualization results into the main paper for better readability?

---

> ### Author Response · Authors · 2025-11-20
> **Responses to Review by Reviewer QCmH**
>
> We thank the reviewer for your valuable time to review our work and provide constructive feedback.  We address all your concerns point by point below.
>
> ---
>
> >**Weakness 1: Cross-model overwriting is more frequent than self-re watermarking for attackers to do in many real-world settings. Protectors , leaking the encoder is not accepted in practical applications. Protectors usually do not use the encoder after it leaks. The motivation or task setting is not reasonable.**
>
> We acknowledge that cross-model overwriting is more frequent than self-re-watermarking due to the presence of different models. Our model can also resist cross-model overwriting attempts while being robust against self-re-watermarking attack. We have included the ability of our model to recover the original watermark when overwritten using other baseline models in Appendix F.4 (page 26) of our revised manuscript.  We have included the table below for your convenience.
>
> | **Model** | **ACC$_{\text{orig}}$** |
> |-----------|------------------------|
> | SSL       | 100.00                 |
> | WFORMER   | 99.35                  |
> | HiDDeN    | 99.23                  |
> | MBRS      | 99.52                  |
> | ARWGAN    | 99.23                  |
> | VINE      | 100.00                 |
>
> Regarding leakage of encoder, our work intentionally considers a white-box adversary with full access to the watermarking models, representing the strongest possible threat model. Access to a leaked encoder through leakage, sharing, or reverse engineering is realistic, as shown by research on model extraction attacks [1]. Furthermore, the LLaMa incident [2] serves as a real-world example of model leakage. When such leakges happen, image owners might continue using the watermarking system, unaware of any model leakage, putting the rights to their valuable images at risk. As the reviewer suggested, it is true that the leaked models are discarded and prevented from re-use. Even if the leaked model is eventually discarded, previously watermarked images remain vulnerable, which can severely undermine trust in watermarking systems.  Motivated by these considerations, our goal is to design a system that remains robust and trustworthy even under worst-case scenarios, rather than relying on the assumption that a model can be perfectly protected indefinitely. We have incorporated this discussion into the Introduction of our revised manuscript. (page 2 Line 73-79).
>
>  [1 ] Adnan Siraj Rakin, Md Hafizul Islam Chowdhuryy, Fan Yao, and Deliang Fan. Deepsteal: Advanced model extractions leveraging efficient weight stealing in memories. In 2022 IEEE Symposium on Security and Privacy (SP), pp. 1157–1174, 2022. doi: 10.1109/SP46214.2022.9833743.
>
> [2] James Vincent. Meta’s powerful ai language model has leaked online — what happens now? The Verge, March 2023. URL https://www.theverge.com/2023/3/8/23629362/ meta-ai-language-model-llama-leak-online-misuse. Accessed: 2025-09-04.

---

> > ### Author Response · Authors · 2025-11-20
> >
> > >**Weakness 2: The defense mechanisms (Lipschitz awareness) are known techniques for bounding sensitivity; the novelty lies more in applying them to watermarking. This is fine but it means the new contribution is primarily the threat framing and integration, not a fundamentally new robustness technique**
> >
> > We acknowledge that  Lipschitz constraints are a well-established tool for bounding the sensitivity background in the paper. However, simply transferring those constraints into a watermarking pipeline is not trivial. Enforcing Lipschitz bounds on the encoder and decoder tightens their sensitivity, hence reducing their effective capacity, creating a direct trade-off between invisibility and recoverability. Addressing that trade-off required an adaptive weighting mechanism to balance competing objectives such as fidelity, robustness to image-processing attacks, and robustness to overwriting attacks. This approach allows the system to preserve visual fidelity while still maintaining strong robustness to both image-processing attacks and re-watermarking attacks.
> >
> > Beyond algorithmic integration, we introduce and validate an operational threat framing that, to our knowledge, is missing from prior work. We demonstrate different ways of malicious re-watermarking that expose a significant vulnerability in existing systems, and we show empirically that our approach mitigates these attacks. In addition, even under constrained model sensitivity, our models achieve a competitive state-of-the-art robustness against standard pixel-based corruptions and a wide range of geometric manipulations, while maintaining acceptable perceptual quality. We also supplement empirical results with theoretical bounds for the rewrite scenario, providing formal insights.
> >
> > Taken together, the contributions of our paper are as follows:
> > - The systematic analysis and attack suite expose an overlooked and severe vulnerability in watermarking systems.
> > - A multiobjective training framework that leverages adaptive objective weighting  to balance the constraints imposed by Lipshcitz constraints and prevent malicious reuse of watermarking systems
> > - Formal analysis on the conditions required for bit flipping and derivation of a bound on the bit error rate under self-re-watermarking attacks. This theoretical analysis is complemented by extensive empirical validation across multiple scenarios to assess and confirm the robustness of our system.
> >
> > We have revised the contributions section of our manuscript (page 2, line 89-107) to more accurately reflect the core advances of our work. We therefore believe the manuscript contributes both new understanding and practical defenses to the watermarking literature.
> >
> >
> >
> > >**Weakness 3: All experiments use 128×128 images and 30-bit payloads. Many watermarking applications use higher resolutions (e.g., 256–1024) and variable payload sizes (32–100+ bits).**
> >
> > We intentionally conducted the majority of our experiments with 128×128 images and a 30-bit payload to maintain consistency with the majority of the established baselines, facilitating direct comparison. To address concerns about higher resolution and larger payloads, we additionally trained a model with 256×256 images embedding 64-bit messages. This model was evaluated under self-re-watermarking, gradient-based attacks, and common image processing perturbations, including JPEG compression, Gaussian blur, crop, cropout, and dropout. The results demonstrate consistent behavior with the 128×128, 30-bit experiments, indicating that our approach generalizes to larger images and higher payloads. These results have been included in Appendix F.3(page 26) of our revised manuscript. Unfortunately, due to memory constraints in our GPU, we were unable to train models at resolutions beyond 256×256. We give the results below for your convenience.
> >  - PSNR 32.72 dB
> > - SSIM 0.98
> > | Attack              | ACC$_{\text{orig}}$ |
> > |--------------------|-------------------|
> > | JPEG (50)           | 99.06             |
> > | Gaussian Blur (2.0) | 99.78             |
> > | Dropout (30%)       | 96.28             |
> > | Cropout (30%)       | 98.24             |
> > | Crop (3.5%)         | 97.63             |
> > | PGD Moderate        | 99.94             |
> > | PGD Strong          | 99.86             |
> >
> >
> >
> > > **Weakness 4: The boxplot can be refined for better visibility, such as setting different intervals for each sub-figure in Fig. 1**
> >
> > We have improved the plot to enhance clarity and visibility in the revised manuscript (page 8).
> >
> > >**Question1: Can you add the framework and visualization results into the main paper for better readability?**
> >
> > We have added the training framework (page 6) and an example of the cover image , watermarked image, and re-watermarked image (Figure 3, page 8) in the main manuscript for better clarity.

---

> ### Comment · Reviewer_QCmH · 2025-11-23
>
> The reviewer appreciates the authors' response but remains unconvinced about the task's motivation. If the watermark encoder has been leaked, the attacker can already use this encoder to claim their ownership—so why they need to re-embed a redundant watermark? In contrast, preventing cross-model overwriting shows higher practicality in protecting our data from being stolen by attackers. The cited blog does not provide sufficient evidence to support the authors' response.

---

> > ### Author Response · Authors · 2025-11-23
> > **Responses to Reviewer QCmH**
> >
> > We thank the reviewer for the follow-up comment and for probing the motivation behind our task setting. Our work focuses on protecting the copyright of existing image assets, rather than on image generation or the encoder itself. Possessing a stolen encoder allows an attacker to generate new images with their own watermark, but this alone does not grant ownership of previously published watermarked images. Consider a cover image $I_a$ watermarked by the owner to produce the corresponding watermarked image with the owner’s watermark $I_w$. An attacker who gains access to both the watermarking system and $I_w$ cannot claim ownership of $I_w$ unless they explicitly re-watermark it with their own watermark. Without this re-watermarking step, ownership verification will correctly attribute the image to the original owner. This distinction motivates our study of self-re-watermarking, which represents a practical threat to ownership verification.
> >
> > As shown in Figure 2 of our manuscript, when the same model is used, **none** of the existing methods can recover the owner’s watermark from the re-watermarked image under self-re-watermarking. In contrast, our methodology resists such malicious reuse, as demonstrated in the same figure. Furthermore, Table 11 (page 26) shows that our model can also withstand cross-watermarking scenarios when tested against several baseline methods.
> >
> > Regarding the cited blog, it was referenced to illustrate a real-world example of a model leak. Multiple other reputable sources have reported the same incident [1], [2]. In our previous response, we also cited [3], which investigates model extraction and its potential to expose a model instance. We further support the possibility of model extraction attacks with the following studies [4], [5], [6].
> >
> > Overall, we strongly believe that self-re-watermarking represents a realistic attack strategy for claiming ownership of valuable images. Our proposed methodology effectively protects against such attacks, as well as against cross-model re-watermarking attempts.
> >
> > We hope this addresses the concerns you may have on this matter and we remain available for further discussion!
> >
> >
> > [1]DeepLearning.AI. (2023, March 15). Runaway LLaMA: How Meta’s LLAMA NLP model leaked. The Batch. https://www.deeplearning.ai/the-batch
> >
> > [2]Center for Security and Emerging Technology. (2023, March 9). Meta’s powerful language models leak online, the U.S. Commerce Department opens up applications for CHIPS funds, and a PRC reorganization will impact R&D and data. https://cset.georgetown.edu/newsletter/march-9-2023/ .
> >
> > [3]  Adnan Siraj Rakin, Md Hafizul Islam Chowdhuryy, Fan Yao, and Deliang Fan. Deepsteal: Advanced model extractions leveraging efficient weight stealing in memories. In 2022 IEEE Symposium on Security and Privacy (SP), pp. 1157–1174, 2022. doi: 10.1109/SP46214.2022.983374
> >
> > [4]Hu, H., & Pang, J. (2021). Stealing Machine Learning Models: Attacks and Countermeasures for Generative Adversarial Networks. In Proceedings of the Annual Computer Security Applications Conference (ACSAC ’21), Virtual Event, USA. ACM. https://doi.org/10.1145/3485832.3485838
> >
> > [5] Y. Gao et al., "DeepTheft: Stealing DNN Model Architectures through Power Side Channel," 2024 IEEE Symposium on Security and Privacy (SP), San Francisco, CA, USA, 2024, pp. 3311-3326, doi: 10.1109/SP54263.2024.00250.
> >
> > [6]Di Mi, Yanjun Zhang, Leo Yu Zhang, Shengshan Hu, Qi Zhong, Haizhuan Yuan, and Shirui Pan. 2024. Towards model extraction attacks in GAN-based image translation via domain shift mitigation. In Proceedings of the Thirty-Eighth AAAI Conference on Artificial Intelligence and Thirty-Sixth Conference on Innovative Applications of Artificial Intelligence and Fourteenth Symposium on Educational Advances in Artificial Intelligence (AAAI'24/IAAI'24/EAAI'24), Vol. 38. AAAI Press, Article 2218, 19902–19910.

---

> > > ### Comment · Reviewer_QCmH · 2025-11-24
> > >
> > > Received with thanks. The reviewer did not raise concerns to the research focus on image asset protection rather than image generation or model protection. The authors pose a fundamental assumption that the watermark model has already been leaked. Under this threat, the extracted ownership information becomes unreliable, as both the legitimate protector and the attacker can extract the embedded watermark. In this situation, the attacker has no incentive to re-embed the watermark into the image. Similarly, the protector is unable to embed watermarks using the proposed method. Although the threat scenario is realistic, the proposed self-re-watermarking concept lacks practical utility under these conditions.

---

> > > > ### Author Response · Authors · 2025-11-24
> > > >
> > > > We are very grateful to the reviewer for further clarifying their concerns and for the time spent on evaluating our work. We address these concerns as follows. First, the reviewer is concerned that the attacker has no incentive to re-embed the watermark into the image in the assumed situation, as both the legitimate owner (or protector) of the image and the attacker can extract the embedded watermark. This issue can be tackled by designing/using a watermark unique to the owner (or protector), e.g., the owner’s logo or name, as it is meaningless for an attacker to reveal the owner’s identifying information. If an attacker intends to claim the image as their own, they would need to embed their own watermark. Second, the reviewer is concerned that the owner (or protector) is unable to embed watermarks using the proposed method. This concern is unnecessary as the owner needs to embed the watermark only once, and there is no need for them to apply a second watermark to an image that already contains their own watermark. In contrast, an attacker would need to apply their watermark to a watermarked image to assert their ownership, but such re-watermarking by an attacker can be prevented by our proposed method, whereas other approaches do not adequately address this problem. We hope these explanations resolve the reviewer’s concerns.

---

> > > > > ### Author Response · Authors · 2025-11-28
> > > > >
> > > > > Dear Reviewer QCmH,
> > > > >
> > > > > Thank you again for your thoughtful comments and follow-up.
> > > > >
> > > > > We have tried our best to address all of your suggestions and concerns. As the deadline for the discussion period is approaching, we would appreciate it if you could review our rebuttal and let us know if any issues remain.
> > > > >
> > > > > Thank you for your time and consideration.
> > > > >
> > > > > Best regards,
> > > > >
> > > > > The Authors

---

### Official Review · Reviewer_RuQL · 2025-10-30

**Soundness:** 3
**Presentation:** 3
**Contribution:** 2
**Rating:** 6
**Confidence:** 3

**Summary:**

This paper introduce a novel deep watermarking framework to defend against this self-re-watermarking attacks.  This attack involves the use of the same encoder to embed a different message into an image that has already been watermarked, which renders the original watermark irretrievable.  Their approach combines Lipschitz constraints on the encoder-decoder sensitivity and re-watermarking adversarial training to limit model sensitivity to input distortions caused by re-embedding. Empirical results demonstrate the new framework’s robustness against self-re-watermarking and various image distortions while maintaining high visual fidelity.

**Strengths:**

1. This paper proposes an innovative solution to address self-re-watermarking attacks. The integration of Lipschitz constraints and adversarial training provides a principled solution, supported by theoretical guarantees.
2. The experiments are thorough and well-designed. they evaluate performance across multiple SOTA baselines, diverse attack scenarios  and standard image distortions. The results consistently show superior robustness while maintaining high visual fidelity.

**Weaknesses:**

1. Self-re-watermarking attacks appear to be resistible through engineered approaches. For instance, one could first verify whether an image has already been embeded watermark (based on decoder perplexity or by training a classifier). If watermarked, the watermark would not be re-embedded during subsequent attempts. Therefore, what advantages does this method offer compared to such approaches?
2. The paper does not analyze composite distortion attacks, such as performing various noise attacks followed by self-re-watermarking.
3. Appendix G demonstrates that re-watermarking results in a significant deterioration of image visual quality, yet the paper does not appear to analyze why this occurs. Furthermore, based on the cases presented in Appendix G, the new artifacts introduced after re-watermarking seem largely similar. Does this indicate limitations in the method?

**Questions:**

Please refer to weaknesses.

---

> ### Author Response · Authors · 2025-11-20
> **Responses to Review by Reviewer RuQL**
>
> We sincerely appreciate the reviewer's constructive feedback. We address all your concerns point by point below.
>
> ---
>
> > **Weakness 1: Self-re-watermarking attacks appear to be resistible through engineered approaches. For instance, one could first verify whether an image has already been embeded watermark (based on decoder perplexity or by training a classifier). If watermarked, the watermark would not be re-embedded during subsequent attempts. Therefore, what advantages does this method offer compared to such approaches?**
>
> Simple engineering checks, such as a classifier or a decoder-perplexity threshold that rejects re-embedding when an image appears watermarked, can work in a black-box setting where the adversary has no access to model internals. Even then, such mechanisms are fragile in stronger black-box threat models. For example, classifier-based checks can be bypassed through black-box evasion attacks [1, 2], and on the other hand, perplexity-based checks are unreliable because decoder robustness is limited to a subset of image distortions.
> Furthermore, if the model is leaked (e.g., the LLaMA incident [3]) or the attacker trains a surrogate encoder[4], the attackers can simply bypass these checks since they have direct access to the watermarking mechanism. For these reasons, the encoder itself must possess an inherent ability to detect and resist re-watermarking attempts, without depending on auxiliary external modules or checks. Our method achieves this by constraining the model’s sensitivity and employing re-watermarking adversarial training, enabling the encoder to internally recognize and suppress re-embedding attempts even in a white box setting where the adversary knows everything about the model. We have revised the Introduction of the manuscript to motivate the necessity of defending against a white-box adversary (page 2, line 73-79).
>
>
> [1] Nicolas Papernot, Patrick McDaniel, Ian Goodfellow, Somesh Jha, Z. Berkay Celik, and Ananthram Swami. 2017. Practical Black-Box Attacks against Machine Learning. In Proceedings of the 2017 ACM on Asia Conference on Computer and Communications Security (ASIA CCS '17). Association for Computing Machinery, New York, NY, USA, 506–519.
>
> [2] Florian Tramèr, Fan Zhang, Ari Juels, Michael K. Reiter, and Thomas Ristenpart. 2016. Stealing machine learning models via prediction APIs. In Proceedings of the 25th USENIX Conference on Security Symposium (SEC'16). USENIX Association, USA, 601–618.
>
> [3] James Vincent. Meta’s powerful ai language model has leaked online — what happens now? The Verge, March 2023. URL https://www.theverge.com/2023/3/8/23629362/ meta-ai-language-model-llama-leak-online-misuse. Accessed: 2025-09-04.
>
> [4] Hu, H., \& Pang, J. (2021). Stealing Machine Learning Models: Attacks and Countermeasures for Generative Adversarial Networks. In Proceedings of the Annual Computer Security Applications Conference (ACSAC ’21), Virtual Event, USA. ACM. https://doi.org/10.1145/3485832.3485838

---

> > ### Author Response · Authors · 2025-11-20
> >
> > > **Weakness 2: The paper does not analyze composite distortion attacks, such as performing various noise attacks followed by self-re-watermarking.**
> >
> > To resist composite attacks, we finetuned our model by modifying our training pipeline so that the encoder receives the perturbed version of the watermarked image as feedback rather than the clean, unperturbed image. We then re-evaluated the updated model on all experiments reported in the paper, including ablation studies and the models developed for model-replication attack. The results from the new model are consistent with those of the previous version, and the manuscript has been updated accordingly. The updated results are shown in Figure 2 (page 8), Figure 4 (page 9), Figure 5 (page 10), Figure 7 (page 18), Figure 8 (page 20),  Figure 9 (page 22), Figure 10 (page 23), Figure 11 (page 24), Figure 12 (page 25), Figure 15 (page 30), Table 1(page 10, line 523), Table 2( page 17), Table 3 (page 19), Table 4 (page 20), Table 5(page 20), Table 7 (page 23), Table 8(page 24), Table 10(page 26), and Table 12 (page 27).
> > Furthermore, as suggested by the reviewer, we conducted additional experiments on re-watermarking after various noise-based distortions and included these results in Appendix E.1 (page 21). The findings demonstrate that the new model is robust to different types of attacks even after image processing. We have included the results table below.
> >
> > | **Image Processing Attack**       | **ACC$_{\text{orig}}$** |
> > |-------------------------------|-------------------|
> > | JPEG (80)                     | 95.09             |
> > | Gaussian Blur (1.0)           | 99.99             |
> > | Cropout (10%)                  | 96.07             |
> > | Dropout (10%)                  | 99.96             |
> > | Gaussian Noise (1.0)           | 100.00            |
> > | Histogram Equalization         | 99.88             |
> > | Crop (3.5%)                    | 99.98             |
> > | Rotate (10°)                   | 94.33             |
> > | Horizontal Flip                | 99.53             |
> > | Vertical Flip                  | 99.45             |
> >
> > >**Weakness 3: Appendix G demonstrates that re-watermarking results in a significant deterioration of image visual quality, yet the paper does not appear to analyze why this occurs. Furthermore, based on the cases presented in Appendix G, the new artifacts introduced after re-watermarking seem largely similar. Does this indicate limitations in the method?**
> >
> > To explain the intrinsic difference between the first and second watermarking stages, we analyzed the signal behavior using Fast Fourier Transform (FFT) and pixel intensity histograms. These analyses reveal that during re-watermarking, the Encoder systematically pushes pixels toward higher-intensity (brighter) regions. We provide a detailed analysis in Appendix G (page 27) of the revised manuscript. Visualizations of these artifacts, along with the cover, watermark, and re-watermarked images, are shown in Appendix I (page 30)  of the revised manuscript. They illustrate the image-dependent nature of these distortions.  We summarize the analysis in Appendix H below for the reviewer's convenience.
> >
> > The introduction of new artifacts after re-watermarking is a direct consequence of our asymmetric training design. The Encoder is optimized to maintain fidelity only during the first watermarking pass. In contrast, the re-watermarking scenario is treated as an adversarial attack where the Encoder is not constrained to preserve visual quality in this step, while the Decoder is explicitly optimized (via Robustness Loss) to survive the resulting distortion. Consequently, the re-watermarking process introduces high-frequency patterns and brightness shifts that effectively act as noise. Because the Decoder is trained on the adversarial examples, it is robust to such high-magnitude distortions and reliably recovers the original watermark.

---

> > > ### Author Response · Authors · 2025-11-24
> > >
> > > Dear Reviewer,
> > >
> > > Thank you once again for your thoughtful comments and valuable feedback.
> > >
> > > We have made every effort to address all of your suggestions and concerns, and we would be grateful if you could review our rebuttal and let us know if any questions or concerns remain.
> > >
> > > Thank you very much for your time and consideration.
> > >
> > > Best regards,
> > >
> > > Authors

---

> > ### Comment · Reviewer_RuQL · 2025-11-27
> >
> > The author addressed weaknesses 2 and 3, but I still have questions about weakness 1. What I'm referring to is that the owner of the watermark encoder can use a classifier to determine whether a watermark has already been embedded in to a image. If it has been embedded, there is no need to embed it again. However, the examples the author  have provided appear to be for black-box attacks. Could you elaborate on this?

---

> > > ### Author Response · Authors · 2025-11-28
> > >
> > > We thank Reviewer RuQL for the follow-up comment. In the context of self-re-watermark attack, the attacker, possessing the encoding module, launches the attack to embed their unique watermark on a previously watermarked image to claim ownership of that watermarked image. Since the attacker has full access to the encoding module (white-box scenario), they can do re-watermarking by running the encoder directly, bypassing any additional checks. In other words, even if the owner can include a classifier in the encoding module to check whether a watermark has already been embedded into an image, this function can be bypassed by the attacker due to their full access to the encoding module. We hope this clarifies the reviewer’s concern.

---

> > > > ### Comment · Reviewer_RuQL · 2025-11-28
> > > >
> > > > Does this imply that the attack can only function in a white-box environment, as the attacker requires full access to the encoding module? However, in real-world scenarios, attackers typically operate solely within black-box environments. For such attackers, watermark embedding can only be achieved through methods like API invocation. Under these circumstances, it appears the attacker cannot bypass additional checks.

---

> > > > > ### Author Response · Authors · 2025-11-28
> > > > >
> > > > > We highly appreciate Reviewer RuQL for the insightful comment and the follow-up question. It is true that in real-world scenarios, attackers can operate under black-box conditions, where the attacker does not have direct access to the watermark encoding model but can interact with it through an API invocation. We agree that an external classifier could be placed before the encoding model to check whether an input image has been watermarked. In principle, this approach can function in a black-box environment, yet it is not consistently reliable because the classifier can be easily fooled [1-2]. For instance, an attacker can introduce mild perturbations or apply light image processing to a watermarked image to mislead the classifier to classify the watermarked image as a non-watermarked image. In contrast, our proposed method does not suffer from this limitation. Furthermore, our method possesses an intrinsic ability to detect re-watermarking attempts and block them without relying on external classifiers or checks. As a result, our method is robust against attacks in both black-box and white-box environments.
> > > > >
> > > > > [1] Nicolas Papernot, Patrick McDaniel, Ian Goodfellow, Somesh Jha, Z. Berkay Celik, and Ananthram Swami. 2017. Practical Black-Box Attacks against Machine Learning. In Proceedings of the 2017 ACM on Asia Conference on Computer and Communications Security (ASIA CCS '17). Association for Computing Machinery, New York, NY, USA, 506–519.
> > > > >
> > > > > [2] Florian Tramèr, Fan Zhang, Ari Juels, Michael K. Reiter, and Thomas Ristenpart. 2016. Stealing machine learning models via prediction APIs. In Proceedings of the 25th USENIX Conference on Security Symposium (SEC'16). USENIX Association, USA, 601–618.

---

### Author Response · Authors · 2025-11-29
**Summary of the Rebuttal Discussions**

We thank all reviewers for their thoughtful feedback and the productive discussions during the rebuttal phase. Below is summary of our discussion with each reviewer.

Reviewer **bosH**: This reviewer initially raised three concerns and four questions. Our detailed rebuttal and the associated updates to the manuscript fully addressed them, and the reviewer expressed satisfaction with the outcome.

Reviewer **RuQL**: The reviewer initially raised three concerns. Our initial response and the revisions made to the manuscript resolved two of them entirely. The follow-up discussion was related to Weakness 1 because the reviewer thought that an engineered approach of using a classifier before the watermarking model can deal with re-watermarking. We clarified that this approach is not reliable because: (1) in the black-box setting, an attacker can introduce mild perturbations or apply light image processing to a watermarked image to mislead the classifier to classify the watermarked image as a non-watermarked image, and (2) in white-box setting, the classifier can be bypassed by the attacker. In contrast, our proposed method is robust against attacks in both black-box and white-box environments.

Reviewer **QCmH**: This reviewer initially listed four concerns and one question. Our initial response and associated updates in the revised manuscript addressed the question and three of those concerns. The remaining discussion centred on the practical implications of model leakage related to Weakness 1. The reviewer’s concern is that if the watermarking model is leaked, the extracted ownership information becomes unreliable, as both the legitimate owner (or protector) and the attacker can extract the embedded watermark. We explained that this issue can be easily tackled by designing/using a watermark unique to the owner (or protector), e.g., the owner’s logo or name, as it is meaningless for an attacker to reveal the owner’s identifying information. If an attacker intends to claim the image as their own, they need to embed their own watermark. We further explained that, unlike prior approaches, our method can effectively prevent this attack.

In summary, we believe we have thoroughly addressed all the concerns raised by the reviewers and revised the manuscript accordingly. We thank the reviewers once again for their constructive role in strengthening our work.

---

### Meta-Review · Area_Chair_Mrrx · 2026-01-13

**Summary:**

This paper introduces self-re-watermarking as a threat model for image watermarking. The authors assumed that an adversary reuses the same encoder to embed a new message into an already-watermarked image which can prevent recovery of the original message without introducing perceptual artifacts. The authors showed that several existing encoder–decoder watermarking methods fail under this attack. The authors proposed a self-aware defense combining Lipschitz-style sensitivity control for the encoder/decoder and re-watermarking adversarial training. Then this claim is supported by a theoretical analysis and empirical evaluation. There are three reviewers for this paper with two positive and one negative. The most positive reviewer explicitly reported no remaining concerns after rebuttal. The other positive reviewer continues to question whether “engineered” pre-checks (e.g., watermark detection/classifier) would suffice in realistic black-box/API settings. The negative reviewer questioned practical motivation/utility of the self-re-watermarking setting under encoder leakage. After reading all rebuttals, the AC thinks this paper can be accepted although some discussions can be further improved in the future works.

**Reviewer Concerns:**

Despite the unfinished rebuttal, the paper (1) identified a concrete and previously under-emphasized failure mode for widely used deep watermarking pipelines, and (2) provided a principled defense with substantial support. There are still some minor concerns that are not fully addressed. The AC would suggest the authors to fix them in the revision or consider deeper research in future.
- **Threat scenarios**. The key issue is whether the attacker capability (malicious reuse of the same encoder, potentially via leakage/white-box access) is central enough to justify the framing, and whether leakage undermines the ownership-verification in practice.
- **Scalability**. The paper adds 256×256/64-bit, but did not demonstrate higher resolutions.

**Reviewer Scores:**

The reviewers engaged in the discussion and the scores are not likely to update any more (one clear positive, one borderline positive, and one borderline negative). But given the rebuttals have addressed most of the questions, the paper can be accepted.

---

### Decision · Program_Chairs · 2026-01-26

Accept (Poster)